# Ubiquitous purine sensor modulates diverse signal transduction pathways in bacteria

Elizabet Monteagudo-Cascales [1,5], Vadim M. Gumerov [2,5], Matilde Fernández [3], Miguel A. Matilla [1], José A. Gavira [4], Igor B. Zhulin [2] ✉ & Tino Krell [1] ✉

Purines and their derivatives control intracellular energy homeostasis and nucleotide synthesis, and act as signaling molecules. Here, we combine structural and sequence information to define a purine-binding motif that is present in sensor domains of thousands of bacterial receptors that modulate motility, gene expression, metabolism, and second-messenger turnover. Microcalorimetric titrations of selected sensor domains validate their ability to specifically bind purine derivatives, and evolutionary analyses indicate that purine sensors share a common ancestor with amino-acid receptors. Furthermore, we provide experimental evidence of physiological relevance of purine sensing in a second-messenger signaling system that modulates c-di-GMP levels.

Bacteria respond to changing environmental conditions by the concerted action of various receptor proteins feeding into multiple signal transduction pathways. Major families of signal transduction proteins include transcriptional regulators, sensor histidine kinases, chemoreceptors, cyclic (di)nucleotide cyclases and phosphodiesterases, serine/threonine protein kinases and phosphatases[1,2]. Binding of a small molecule ligand to a sensor domain (also called a ligand binding domain, LBD) is a canonical mode of signal recognition (sensing) by these receptors. The same type of sensor domain is frequently found in different types of signal transduction proteins[3–5], indicating that sensor domains have been exchanged and recombined with various receptors during evolution. Although hundreds of different sensor domains have evolved[2,6] and new ones are regularly being discovered[7,8], only a few of them are ubiquitous. PAS (Per-Arnt-Sim) and Cache (Calcium channels-chemotaxis) domains form two large superfamilies of intracellular and extracellular sensors, respectively, that are found in bacterial, archaeal and eukaryotic signal transduction systems[9–11].

Signals that are recognized by most bacterial receptors are unknown, presenting a major challenge in microbiological research[12].

Obtaining such knowledge is critical for understanding the physiological role of signal transduction pathways and development of approaches to fight bacterial virulence by interfering with signaling. However, sensor domains evolve extremely fast, leading to a large degree of sequence divergence[13,14], which hinders predicting signaling molecules for millions of unstudied receptors by extrapolation from a few experimentally characterized homologs.

We have recently reported the successful prediction and verification of signaling molecules for two large sensor domain families[11,15]. By combining sequence and structure information from several experimentally characterized Cache domains, we derived ligand recognizing motifs that were used to computationally identify thousands of motif-containing homologs and to experimentally confirm predicted ligands in selected targets. Subsequently, we defined two large subfamilies of amino acid-sensing (termed dCache_1AA)[11] and biogenic amine-sensing (termed dCache_1AM)[15] domains.

Purines are central intermediates in the synthesis of nucleotides and nucleic acids and are among the most abundant metabolites of living organisms[16]. Purine metabolites also provide cells with the necessary energy and cofactors to promote cell survival and

[1]Department of Biotechnology and Environmental Protection, Estación Experimental del Zaidín, Consejo Superior de Investigaciones Científicas, Prof. Albareda 1, 18008 Granada, Spain. [2]Department of Microbiology and Translational Data Analytics Institute, The Ohio State University, Columbus, OH 43210, USA. [3]Department of Microbiology, Facultad de Farmacia, Campus Universitario de Cartuja, Universidad de Granada, 18071 Granada, Spain. [4]Laboratory of Crystallographic Studies (CSIC-UGR), Avenida de las Palmeras 4, 18100 Armilla, Spain. [5]These authors contributed equally: Elizabet Monteagudo-Cascales, Vadim M. Gumerov. ✉e-mail: jouline.1@osu.edu; tino.krell@eez.csic.es

proliferation[16,17]. In addition, purines act as signaling molecules in eukaryotes to coordinate multiple cell behaviors and physiological processes[18]. There is also evidence that purines act as signal molecules in bacteria[19–21].

We have previously identified the chemoreceptor McpH as the first bacterial receptor that exclusively recognizes purine derivatives via its dCache_1 domain mediating chemoattraction to these compounds[22]. In this study, we used computational and experimental approaches to identify sensors that specifically bind purines and, with lower affinity, pyrimidines. By combining analysis of the newly solved 3D structure of the McpH-LBD in complex with uric acid and comparative protein sequence analysis, we identify a sequence motif for purine binding in thousands of dCache_1 domains (termed dCache_1PU) deposited in public databases. We experimentally verify this computational prediction in selected targets, all of which were found to bind purine and, with a lower affinity, pyrimidine derivatives. We demonstrate the physiological role of purine sensing in a second messenger signaling system. This study further demonstrates that our approach is applicable to defining ligand binding domain families and reveals roles of purine derivatives as important extracellular signals in bacteria.

## Results

### Structural basis for ligand recognition at McpH-LBD

The individual McpH-LBD was generated as a his-tagged fusion protein, purified, and crystallized in complex with uric acid. No crystals were obtained for the apo protein. The 3D structure of McpH-LBD was solved by molecular replacement using an AlphaFold2[23] model to a resolution of 1.95 Å (Supplementary Fig. 1). McpH-LBD adopts a structure typically observed for dCache_1 domains that is composed of a long C-terminal helix and two α/β type PAS folds that are termed membrane-distal and membrane-proximal modules (Fig. 1).

Structural alignments with all currently deposited structures in the protein data bank[24] show that McpH-LBD shares significant structural similarity with LBDs that bind quaternary amines (Supplementary Table 1). Well-defined electron density was obtained for bound uric acid that enabled to precisely introduce the ligand model (Fig. 2A). Similarly to most other dCache_1 domains[24], the ligand is bound to the membrane distal module (Fig. 1). The inspection of the molecular detail of ligand recognition shows that uric acid is sandwiched between two aromatic residues, W140 and F167, establishing π-stacking interactions (Fig. 2B). Furthermore, three of the four N-atoms of the purine backbone establish a hydrogen bond involving amino acids D156, D169 and Y121 (Fig. 2B and Supplementary Fig. 2). Two of

the three uric acid oxygen atoms, that are not part of the purine skeleton, establish three hydrogens bonds with R129, N154 and G159 (Fig. 2B and Supplementary Fig. 2). In interpreting these binding interactions it has to be taken into account that the protonation state of the N7 atom of uric acid is unclear at present.

### A conserved sequence motif for purine binding in dCache_1 domains

To identify the molecular determinants of purine binding and elucidate the prevalence of purine binding receptors, we performed searches against the NCBI RefSeq database using the MpcH dCache_1 domain (see "Methods") and collected over 6000 homologous dCache_1 domain sequences. Next, we built a multiple sequence alignment using collected sequences and tracked residues forming the ligand binding interface in the McpH dCache_1 domain. Out of ten residues forming contacts with the ligand in McpH, five showed consistent conservation across all sequences collected (Fig. 2C). We defined the combination of these residues as a purine binding motif (Fig. 2D) and a group of dCache_1 domains containing this motif as dCache_1PU. This domain has been identified in about 6,300 proteins that belong to all major types of bacterial transmembrane receptors: chemoreceptors, sensor histidine kinases, serine/threonine phosphatases as well as diguanylate cyclases and phosphodiesterases from numerous bacterial species (Supplementary Data 1 and Supplementary Fig. 4). A significant number of these proteins are encoded in the genomes of species belonging to genera frequently associated with disease in humans/animals (e.g., *Aeromonas, Campylobacter, Burkholderia, Clostridioides, Vibrio*) or plants (e.g., *Burkholderia, Dickeya, Pectobacterium*). Other species are mainly free-living bacteria that have been isolated from a diverse range of ecological niches (Supplementary Data 2). dCache_1PU appears to be confined to the bacterial kingdom as no such homologs have been identified in archaea or eukaryotes.

### Experimental validation of the computationally derived purine binding motif

To assess the contribution of the individual amino acids of the purine binding motif, native McpH-LBD and the corresponding five alanine substitution mutants were purified and submitted to microcalorimetric titrations with adenine. Since in sequence alignments there is frequently an asparagine at the place of D169, we have also analyzed the D169N mutant. Figure 3 shows the individual binding curves that have been generated using the same titration protocol as used for analyzing the wild-type domain.

Since mutant protein showed very significant reductions in the binding signal, titrations were repeated with the highest possible ligand concentration (Supplementary Fig. 5) to derive the binding constants (Table 1). Analyses show that D169 is an essential amino acid in McpH for binding since its replacement with Ala abolished binding, whereas the D169N mutant bound adenine with an 80-fold reduced affinity. However, the majority of homologous receptors have N at this position and our subsequent experiments showed that they bind purine and pyrimidine compounds. These results may point to the fine tuning of the receptors and that some compounds may bind more tightly with N instead of D at this position. The mutation of the amino acids that maintain the π-stacking interactions with the bound ligand, W140 and F167, resulted in a 65 and 47-fold reduction in affinity, respectively. A similarly important reduction was observed for the Y121A mutant, whereas the most modest effect on affinity, a 16-fold reduction, was recorded for the R129A mutant (Table 1). R129 established a hydrogen bond with the carbonyl group at position 8 of bound uric acid (Fig. 2B). Since other McpH ligands do not possess a carbonyl group at this position, we have conducted docking experiments of further McpH ligands to the protein structure showing that R129 establishes hydrogen bonds with other parts of the bound ligand (Supplementary Fig. 6).

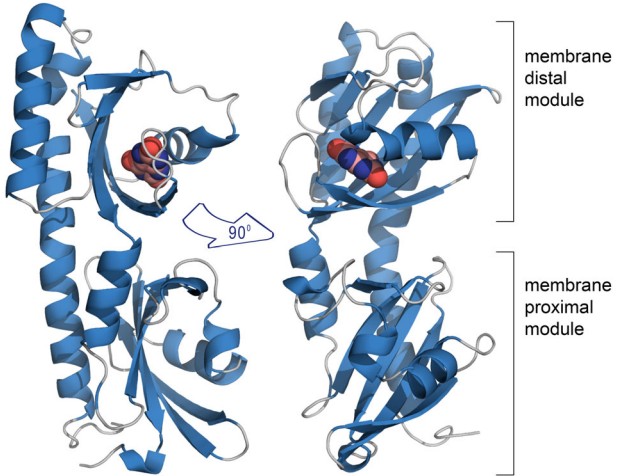

**Fig. 1 | The overall structure of McpH-LBD in complex with uric acid.** Ribbon diagram with bound uric acid shown in spacefill mode.

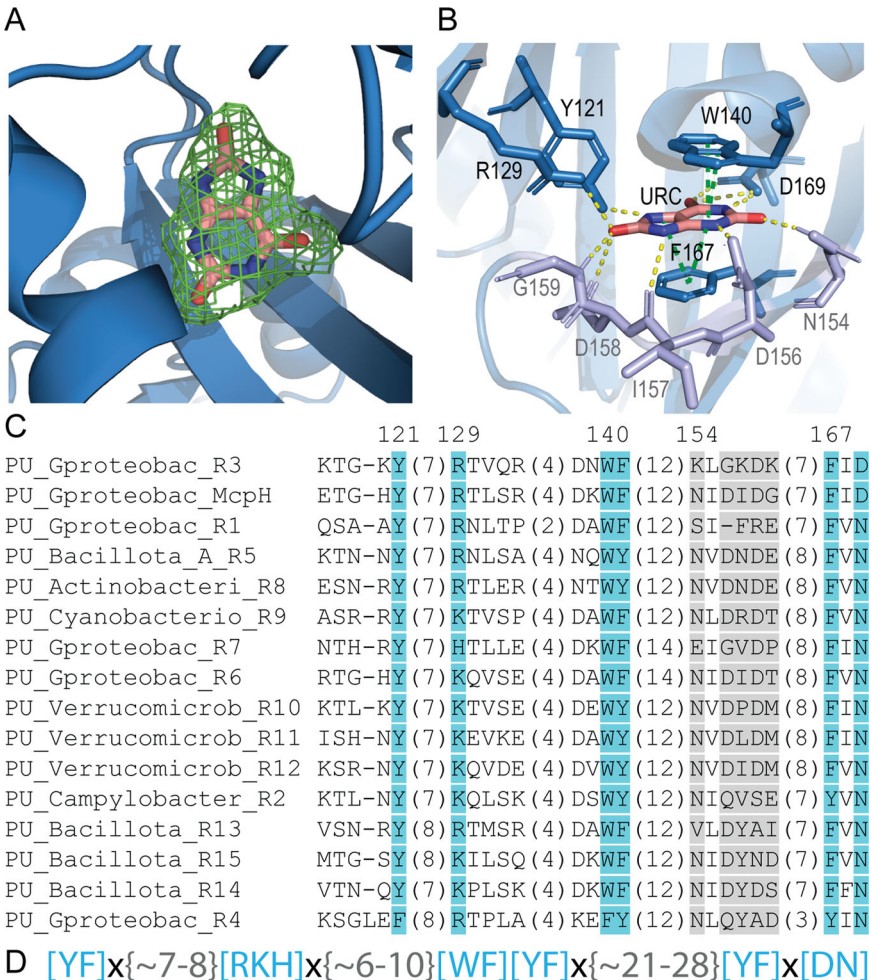

**Fig. 2 | Computationally derived purine binding motif. A, B** The molecular detail of uric acid recognition at McpH-LBD. **A** The mesh corresponds to the initial |Fo-Fc| electron density map containing bound uric acid contoured at 1.5 σ level. The uric acid model placed into this density is shown in stick mode. **B** Ligand binding pocket of the McpH dCache_1 domain. Residues constituting the purine binding motif are shown in blue, variable residues involved in ligand binding are shown in light purple. Contacts in yellow−hydrogen bonds; contacts in green−pi-stacking; URC− uric acid. Molecular interactions were identified by Mol* Viewer using a cut-off of 3.5 Å[72]. **C** Multiple sequence alignment showing the predicted purine binding motif (positions in blue as well as an aromatic position corresponding to F141 in McpH). Positions shown in light gray form contacts with the ligand but are not conserved across the full alignment. Phylum/class names are provided. Endings R1, R2, etc. in protein names refer to the proteins analyzed in this study (Table 2). A full alignment can be found in Supplementary Fig. 3. **D** The purine binding motif. Residues in brackets indicate that any of the specified residue can be at the corresponding position. "X" denotes any residue, numbers in curly brackets indicate the distance in amino acids between the specified positions.

## Computationally identified dCache_1PU domains specifically bind purine and pyrimidine derivatives

To verify the computational identification of purine-responsive dCache_1 domains, we have selected 15 receptors using several criteria: (1) purine binding motif variant; (2) taxonomic diversity; (3) receptor type; and (4) species pathogenicity status (Table 2, termed R1 to R15). These 15 proteins were expressed in *Escherichia coli* and 11 of them were obtained as stable soluble proteins permitting their analysis. These proteins were submitted to thermal shift assays using the Biolog compound arrays PM3B (nitrogen sources) and PM5 (nutrient supplements). These arrays contain a broad selection of different purines, pyrimidines, polyamines, quaternary amines, sugars, amino acids, and organic acids. In addition, proteins were screened against a series of other purine compounds. Thermal shift assays monitor the change in the midpoint of protein unfolding (Tm) induced by ligand binding. Increases in Tm above 2 °C are considered significant, in which cases the corresponding dissociation constants ($K_D$) were derived from microcalorimetric titrations (Fig. 4). No compounds other than those listed in Table 3 caused a Tm shift superior to 2 °C.

Due to the limitations imposed by ligand dilution heats, ITC is suitable to study high-affinity binding events. In a number of cases, very significant increases in Tm were observed, but microcalorimetric binding heats were very weak (not permitting data analysis), indicative of low affinity binding. Binding data are summarized in Table 3 from which a number of conclusions can be derived. (1) Evidence of purine binding was obtained for all 11 proteins that could be generated as soluble recombinant protein, indicating that bioinformatic predictions are highly precise (Fig. 4 and Supplementary Fig. 7). (2) For five proteins, significant increases in Tm were observed for the pyrimidines thymine and cytosine and the corresponding $K_D$ values were obtained in ITC. This indicates that this domain family is specific for purines and pyrimidines. Purines are composed of a pyrimidine and imidazole moiety. Ligand docking studies to a R4 model showed that bound pyrimidines overlapped with the pyrimidine moiety of theophylline (Supplementary Fig. 8). The structural resemblance between pyrimidines and part of the purine structure is thus likely the reason for the pyrimidine recognition by some family members. However, none of the pyrimidines bound with low $K_D$ (below 10 μM), indicative of preferential purine recognition. (3) Theophylline, a bi-methyl

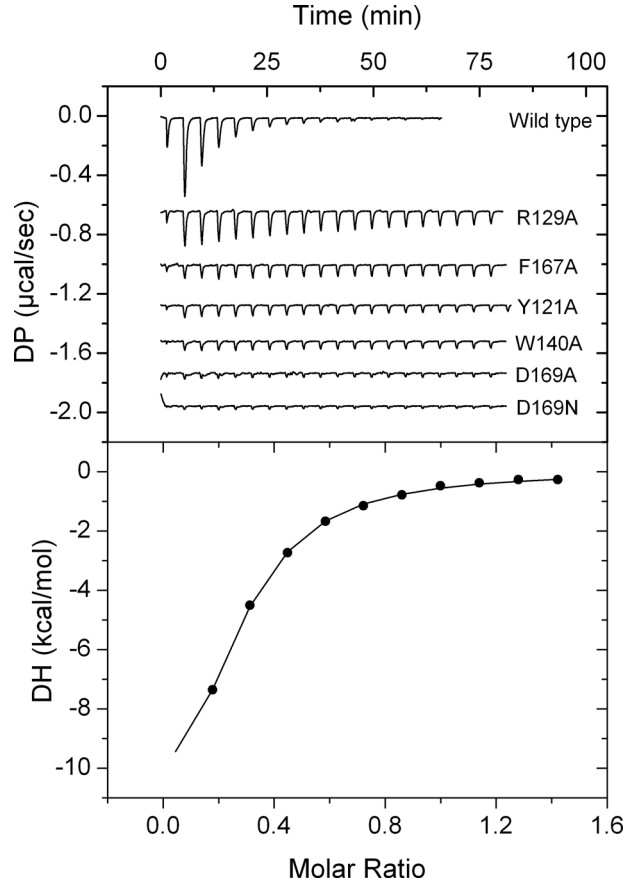

**Fig. 3 | Assessment of the contribution of motif amino acids to binding.**
Microcalorimetric titrations of McpH-LBD and site-directed mutants with adenine.
Upper panel: heat changes for the titration of 9–18 µM protein with 12.8 µl aliquots
of 300 µM adenine. Lower panel: integrated, dilution heat corrected and con-
centration normalized titration data of the wild-type protein fitted with the "One
binding site model" of the MicroCal version of ORIGIN. Titrations of mutant pro-
teins were repeated with the maximal possible adenine concentration (Supple-
mentary Fig. 5) to derive, if possible, the binding parameters.

**Table 1 | Dissociation constants ($K_D$) derived from the
microcalorimetric titration of McpH-LBD and site-directed
mutants in amino acids of the purine binding motif with
adenine**

| Protein | $K_D$ (µM) | Fold reduction to wild-type protein |
|---|---|---|
| McpH-LBD | 2.6 ± 0.4 | – |
| McpH-LBD Y121A | 151 ± 7 | 58 |
| McpH-LBD R129A | 42 ± 2 | 16 |
| McpH-LBD W140A | 169 ± 66 | 65 |
| McpH-LBD F167A | 121 ± 30 | 47 |
| McpH-LBD D169A | No binding | – |
| McpH-LBD D169N | 209 ± 45 | 80 |

substituted xanthine derivative, was the only ligand that bound to all
proteins analyzed. In contrast, only 2 proteins recognized xanthine
and none of the proteins recognized caffeine, which is N7 methyl-
substituted theophylline (Supplementary Fig. 9). This shows that
bimethylation of xanthine (theophylline) is a frequent requirement of
ligand binding, whereas the additional methylation at the N7 of
theophylline, giving rise to caffeine, abolishes recognition. These data
point to an important signaling function of theophylline. (4) There
was only a single receptor that showed a pronounced signal

**Table 2 | Proteins predicted to bind purines and analyzed in this study**

| Name | Ref. Seq ID | Receptor type[a] | Motif variant[b] | Species/strain | Phylogenetic category (Phylum) | Pathogenicity/characteristics[c] |
|---|---|---|---|---|---|---|
| R1 | WP_219614703 | CR | YRWFN | *Aeromonas salmonicida* | Pseudomonadota | Fish pathogen, mainly salmon. Important economic losses |
| R2 | WP_107944080 | CR | YKWYN | *Campylobacter concisus* | Campylobacterota | Human pathogen, causes inflammatory bowel disease |
| R3 | WP_124259872 | CR | YRWFD | *Burkholderia vietnamiensis* | Pseudomonadota | Human pathogen, causes lung infections. High mortality rate |
| R4 | WP_199478134 | CR | FRFYN | *Marinomonas spartinae* | Pseudomonadota | Halophyte plant isolate, can promote plant growth |
| R5 | WP_131005693 | GGDEF/EAL | YRWFN | *Clostridioides difficile* | Bacillota | Human pathogen, at times fatal. Disrupts gut microbiota |
| R6 | WP_185834821 | GGDEF | YKWFN | *Vibrio cholerae* | Pseudomonadota | Human pathogen, causes life-threatening diarrheal disease |
| R7 | WP_038903150 | GGDEF | YHWFN | *Dickeya zeae* | Pseudomonadota | Plant pathogen, in particular maize and rice |
| R8 | WP_227113344 | GGDEF | YRWFN | *Eggerthella sinensis* | Actinomycetota | Human pathogen, causes life-threatening bacteremia |
| R9 | WP_198537540 | GGDEF | YKWFN | *Vulcanococcus limneticus* | Cyanobacteriota | N₂-fixing bacterium, isolated from freshwater |
| R10 | WP_110129388 | HK | YKWFN | *Coraliomargarita sinensis* | Verrucomicrobiota | Isolated from a marine saltern |
| R11 | WP_162024566 | HK | YKWFN | *Lentimonas* sp. CC4 | Verrucomicrobiota | Efficient degrader of fucoidan, a major algae product |
| R12 | WP_220621668 | HK | YKWFN | *Ruficoccus* sp. ZRK36 | Verrucomicrobiota | Isolated from deep sea cold seep |
| R13 | WP_167859577 | S/T phos | YKWFN | *Paenibacillus cymbidii* | Bacillota | N₂-fixing isolated from orchid roots |
| R14 | WP_021170906 | S/T phos | YKWFN | *Sporomusa ovata* | Bacillota | Anaerobic, endospore forming bacterium, isolated from leaves |
| R15 | WP_207952809 | S/T phos | YRWFN | *Paenibacillus* sp. S3NO8 | Bacillota | Isolated from agricultural soil |

The first column specifies the number given to the different receptors.
[a]CR chemoreceptor, GGDEF/EAL diguanylate cyclase/phosphodiesterase, HK histidine kinase, S/T phos serine/threonine phosphatase.
[b]Ligand binding residues of the dCache_1PU motif are shown. Strands of non-conserved residues between the motif residues are implied according to the motif definition in Fig. 2D.
[c]The corresponding references are provided in Supplementary Table 2.

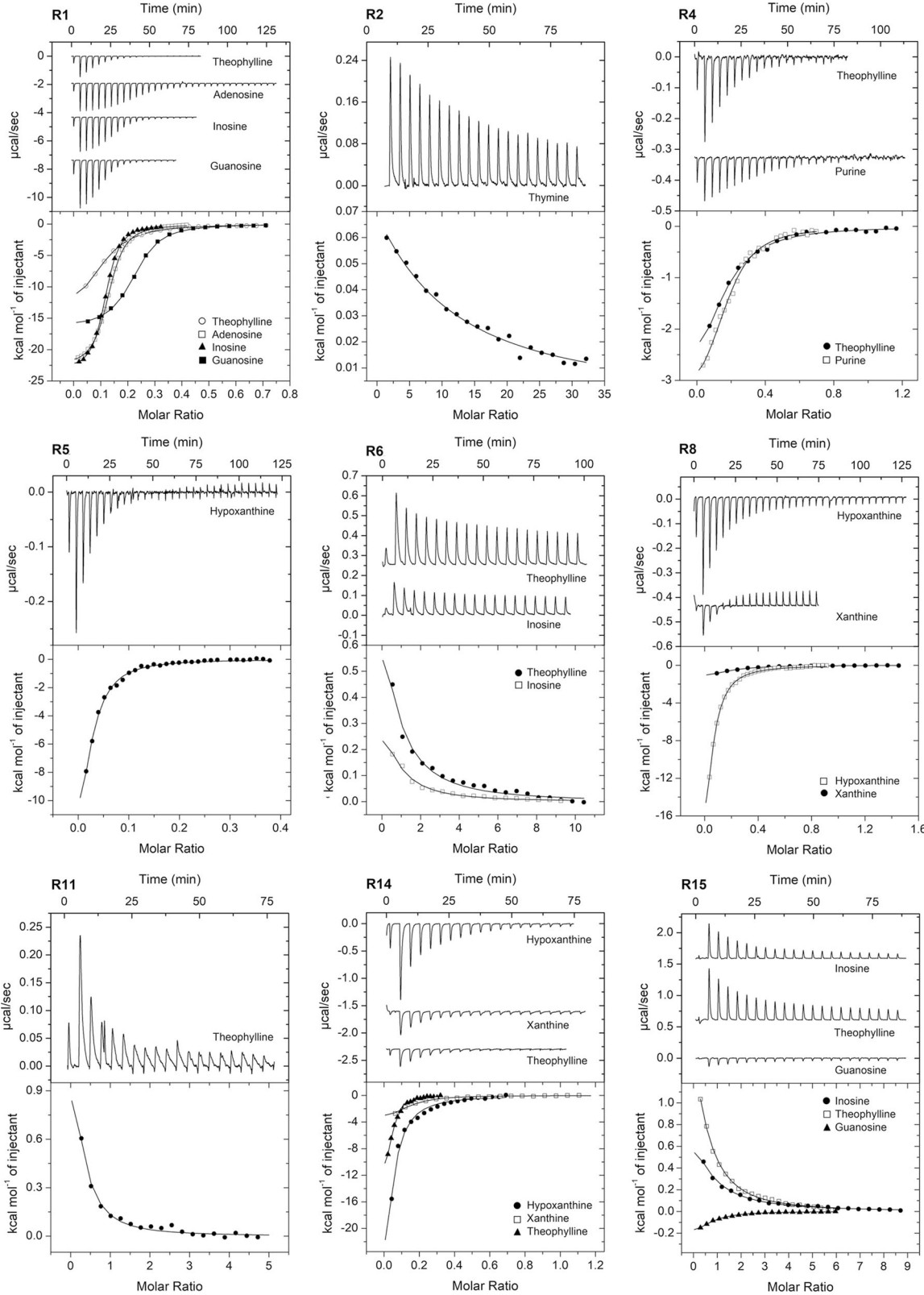

**Fig. 4 | Binding studies of purine derivatives to dCache_1PU domains from various receptor families.** Microcalorimetric titration of 45–138 µM protein with 0.2–20 mM solutions of purine and pyrimidine derivatives. Upper panels: raw titration data. Lower panels: integrated, dilution heat corrected and concentration normalized raw data. The lines are the best fits using the "One binding site" model of the MicroCal version of ORIGIN. The derived dissociation constants are provided in Table 3.

**Table 3 | Dissociation constants derived from microcalorimetric titrations of the LBDs from selected receptors with different ligands**

| Name | Adenine (Purine compounds) | Adenosine | Purine | Guanosine | Guanine | Inosine | Xanthine | Hypoxanthine | Uric acid | Theophylline | Caffeine | Cytosine (Pyrimidine compounds) | Thymine | Allantoin |
|---|---|---|---|---|---|---|---|---|---|---|---|---|---|---|
| R1 | 79 ± 3 | **1.1 ± 0.1** | Nts[b] | **1.2 ± 0.1** | Nts[b] | **0.9 ± 0.1** | Nb[a] | Nts[b] | Nts[b] | **4.2 ± 0.1** | Nts[b] | 1,457 ± 190 | 595 ± 47 | Nts[b] |
| R2 | Nb[a] | Nb[a] | Nts[b] | Nb[a] | Nts[b] | Nb[a] | Nts[b] | 24 ± 3.6 | Nb[a] | **9.6 ± 3.8** | Nb[a] | 52 ± 6.4 | 1,623 ± 510 | Nts[b] |
| R3 | Insoluble protein | | | | | | | | | | | | | |
| R4 | 12 ± 1.4 | 88 ± 12 | **2.3 ± 0.3** | Nts[b] | Nts[b] | Nts[b] | Nts[b] | 561 ± 190 | Nts[b] | **6.2 ± 0.7** | Nb[a] | 74 ± 15 | 21 ± 0.4 | Nts[b] |
| R5 | Nb[a] | Nts[b] | Nb[a] | Nts[b] | Nts[b] | Nts[b] | Nb[a] | **1.6 ± 0.1** | Nb[a] | 214 ± 86 | Nts[b] | Nts[b] | Nts[b] | Nts[b] |
| R6 | Nts[b] | Nts[b] | Nts[b] | 25 ± 5.2 | Nts[b] | 83 ± 19 | Nts[b] | Nb[a] | Nts[b] | 108 ± 32 | Nts[b] | Nts[b] | Nts[b] | Nts[b] |
| R7 | Insoluble protein | | | | | | | | | | | | | |
| R8 | Nts[b] | Nts[b] | Nts[b] | Nts[b] | Nts[b] | 164 ± 23 | **10 ± 1.1** | **2.0 ± 0.1** | Nts[b] | 87 ± 8.7 | Nts[b] | Nts[b] | Nts[b] | Nb[a] |
| R9 | Insufficient protein expression | | | | | | | | | | | | | |
| R10 | | | | Tsa[c] | Nts[b] | Tsa[c] | Tsa[c] | Tsa[c] | | Tsa[c] | | | | |
| R11 | 72 ± 13 | Nb[a] | Nb[a] | 25 ± 4.9 | Nts[b] | **7.7 ± 1.5** | Nb[a] | Nb[a] | Nb[a] | 30 ± 8.1 | Nb[a] | Nb[a] | 854 ± 254 | Nb[a] |
| R12 | Unfolded protein | | | | | | | | | | | | | |
| R13 | Nts[b] | Nts[b] | Nts[b] | Nts[b] | Tsa[c] | Nts[b] | Nts[b] | Nts[b] | Nts[b] | Nts[b] | Nts[b] | Nts[b] | Nts[b] | Nts[b] |
| R14 | Nb[a] | 17 ± 4.1 | 94 ± 19 | 25 ± 4.9 | Nts[b] | **3.5 ± 1.4** | **5.1 ± 0.9** | **6.0 ± 1.5** | Nts[b] | **1.4 ± 0.1** | Nts[b] | 105 ± 20 | 32 ± 8.9 | Nb[a] |
| R15 | Nts[b] | Nb[a] | Nb[a] | 37 ± 4 | Nts[b] | 120 ± 15 | **7.6 ± 1.1** | **1.1 ± 0.1** | Nb[a] | 65 ± 6 | Nb[a] | Nb[a] | Nb[a] | Nts[b] |

Dissociation constants of 10 µM and below, indicative of high-affinity binding, are shown in bold face. Representative titration curves are shown in Fig. 4. The corresponding n-values are provided in Supplementary Table 3.
[a]Nb: no binding observed in microcalorimetric titrations.
[b]Nts: no binding observed in thermal shift assays, i.e., compounds caused changes in the midpoint of protein unfolding (Tm) of less than 2 °C.
[c]Tsa: binding observed in thermal shift assays, i.e., compounds caused changes in Tm of more than 2 °C (Supplementary Fig. 7).

specificity, namely R5 for hypoxanthine. The remaining receptors were characterized by a ligand promiscuity, and three receptors recognized 7 to 9 different ligands. The widest ligand range had R14 that recognized 9 compounds of which 4 with $K_D$ values below 10 µM, indicative of tight binding. (5) A previous analysis of a large number of ligand-sensor domain interactions has revealed that in 60% of the cases the $K_D$ values were in between 1 to 50 µM[25]. The same percentage was obtained for data shown in Table 3, indicating that the signal affinities of this domain family are in the range typically observed for sensor domains. (6) Domains with the same motif variant differed in their ligand profiles (like R11 and R14, Table 2). These differences may be due to changes in the relative position of binding motif residues (Supplementary Fig. 10).

To define the ligand specificity of dCache_1PU family members, we have titrated domains R4, R8 and R14 with non-purinergic compounds. These proteins have been selected since they form part of different receptor families, belong to phylogenetically distant species and differ in the variant of the purine binding motif (Table 2). Each protein was titrated with 15 amino acids and amines that were found to bind frequently to members of the amino acid[11] and amine specific[15] dCache_1 domains. As shown in Supplementary Fig. 11, in all cases the peaks were small and uniform, indicative of an absence of binding of non-purinergic signals to members of the dCache_1PU family.

## Purine sensors share a common ancestor with amino acid sensors

To determine the evolutionary origin of the purine binding sensor domains, we analyzed the dCache_1PU domain from the perspectives of sequence conservation, structure, and phylogeny. Sequence comparison of dCache_1PU with previously studied amino acid (dCache_1AA)[11] and amine (dCache_1AM) binding domains[15] showed that dCache_1PU is slightly more similar to dCache_1AA than to dCache_1AM: (1) while the amine binding domain has an insertion upstream of the ligand binding pocket[15], dCache_1PU along with dCache_1AA does not; and (2) the number of residues inside the dCache_1PU pocket, in the region corresponding to the part of the dCache_1AM domain pocket after the insertion, is on average similar to the number of residues in the dCache_1AA domain, while dCache_1AM exhibits more variability (Fig. 5A and Supplementary Fig. 12). However, the multiple sequence alignment analysis showed that purine receptors have a unique ligand binding motif: the only conserved positions shared with the amino acid and amine motifs correspond to two highly conserved aromatic residues (corresponding to W140 and F141 in McpH in Fig. 5A), with the second one being not directly involved in ligand binding but rather playing an important structural role (Fig. 5B).

To investigate structural similarities, we next superimposed ligand binding pockets of the dCache_1PU domain of McpH from *Pseudomonas putida* (PDB ID 8BMV) with dCache_1AA of PctA from *Pseudomonas aeruginosa* (PDB ID 5T65) and dCache_1AM of McpX from *Sinorhizobium meliloti* (PDB ID 6D8V). Comparison with dCache_1AA showed that the two mentioned aromatic residues ($F141_{McpH}$/$Y129_{PctA}$ and $W140_{McpH}$/$W128_{PctA}$) are in similar positions and have similar orientations in both dCache_1PU and dCache_1AA. The critical amino acid motif residue D173 in PctA corresponds to G185 in McpH, which does not bind ligands and is not part of the purine motif. Another amino acid motif residue in PctA, Y144, in terms of orientation better corresponds to D156 in McpH (Fig. 5B–D), even though the multiple sequence alignment showed that Y144 aligns with I155, which precedes D156 and oriented away from the pocket (Fig. 5A). Y121 of PctA loosely corresponds structurally to Y121/R129 in McpH (Fig. 5B–D)—two important residues for purine binding. Conversely, two purine motif residues, F167 and D169, correspond in PctA to T155 and A157, respectively, two residues that are not part of the amino acid-binding motif (Fig. 5C).

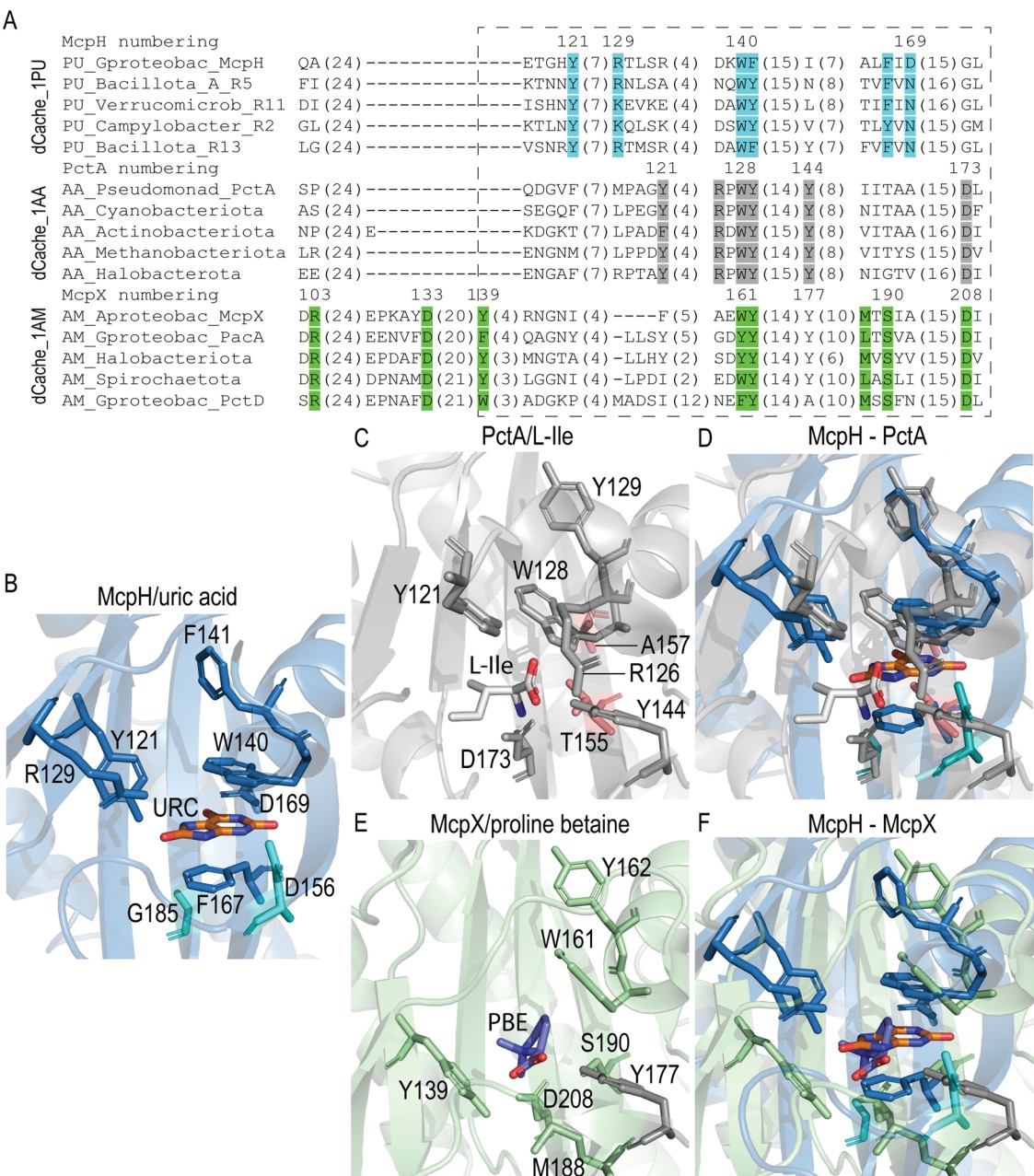

**Fig. 5 | Comparison of dCache_1PU, dCache_1AA, and dCache_1AM sensors.**
**A** Multiple protein sequence alignment of purine, amino acid, and amine binding dCache_1 domains. The corresponding motifs are shown in blue (purine motif), gray (amino acid motif), and green (amine motif). A dashed rectangle denotes a sequence region corresponding to the ligand binding site. A full alignment can be found in Supplementary Fig. 3. **B** Ligand binding pocket of the purine receptor McpH from *Pseudomonas putida*. The two residues in turquoise are in positions corresponding to the amino acid as well as amine motif residues but are not part of the purine motif. URC−uric acid. **C** Ligand binding pocket of the amino acid receptor PctA from *Pseudomonas aeruginosa* with bound L-isoleucine. The two residues in red are in positions corresponding to the purine motif residues but they are not part of the amino acid motif. **D** Structural superimposition of the ligand binding pockets of McpH and PctA. **E** Ligand binding pocket of the amine receptor McpX from *Sinorhizobium meliloti* in complex with proline betaine (PBE). **F** Structural superimposition of the ligand binding pockets of McpH and McpX.

Comparison with dCache_1AM domain showed again that the two aromatic residues ($F141_{McpH}$/$Y162_{McpX}$ and $W140_{McpH}$/$W161_{McpX}$) are in similar positions and have similar orientation in both dCache_1PU and dCache_1AM (Fig. 5B, E, F). The amine receptor residues Y177 and D208 of MpcX correspond to the already mentioned D156 and G185 in McpH, respectively (Fig. 5B, E, F). Interestingly, multiple sequence alignments showed that the position corresponding to D173/D208 in PctA/McpX is strictly conserved in both amino acid and amine motifs, while a conserved aromatic position corresponding to Y121 in PctA is relatively conserved only in McpX (Fig. 5A). The purine motif position F167

corresponds to the amine motif position S190. On the other hand, the dCache_1PU positions R129 and Y121 do not have counterparts in dCache_1AM (Fig. 5B, E, F). Similarly, the dCache_1AM positions Y139 and M188 do not have counterparts in dCache_1PU (Fig. 5B, E, F).

Sequence and structure analyses demonstrated that purine receptors have a unique ligand binding motif. To further elucidate the phyletic relationship between the purine, amino acid and amine receptors, we performed a phylogenetic analysis using protein sequences of the dCache_1 domain of all three receptor families from major phyla, in which they occur (see "Methods"). Previous work has

shown that the amine binding domain is derived from the amino acid-binding domain[15]. Our present analysis using Bayesian phylogenetic inference confirmed this and also showed that the purine binding domain most likely originated from the common ancestor that gave rise to both the amino acid and purine binding domains (Fig. 6 and Supplementary Fig. 13). This conclusion is concordant with the results

of the above sequence and structure analyses (Fig. 5), which revealed the unique features of the purine receptors compared to both amino acid and amine receptors.

## Signals identified modulate the activity of a diguanylate cyclase from *Vibrio cholerae*

*Vibrio cholerae* is among the most important human pathogens that cost the lives of about 100,000 people annually[26]. This species is also a model system to investigate c-di-GMP signaling, controlling the switch between sessile and planktonic lifestyle, which is a feature closely linked to multiple attributes of *V. cholerae* virulence[27,28]. It was shown that c-di-GMP represses *V. cholerae* virulence[28]. The control of c-di-GMP homeostasis is highly complex since *V. cholerae* has 62 diguanylate cyclases and phosphodiesterases, including 31 GGDEF domain containing diguanylate cyclases[28]. However, information on the signals recognized by these receptors is very scarce.

R6 is one of the GGDEF domain containing receptors (VC2224 in *V. cholerae* O1 El Tor). The *VC2224* gene was upregulated in a rugose variant, which may partially account for the rugose phenotype observed[29]. Furthermore, *VC2224* was the only gene involved in c-di-GMP homeostasis that was expressed in vitro and repressed in the intestine during mouse infection[30]. We have shown that the R6 sensor domain recognizes guanosine, inosine and theophylline (Table 3). To assess the effect of these ligands on the activity of the R6 diguanylate cyclase, the *VC2224* gene was heterologously expressed from a pBBR-based plasmid in *P. putida* containing the c-di-GMP bioreporter plasmid pCdrA::*gfp*^C, which allows to quantify c-di-GMP levels using fluorescence. *P. putida* was chosen due to its very low basal c-di-GMP levels[31]. In this species, the development of a wrinkly colony morphology was found to be a consequence of high c-di-GMP levels[32].

We first measured colony morphology and fluorescence on minimal medium agar plates containing either no or increasing concentrations of inosine, theophylline and guanosine (Fig. 7 and Supplementary Fig. 14). Whereas colonies were smooth at inosine concentration of up to 10 μM, colonies were wrinkly from 50 μM onwards (Fig. 7A). This phenotype supported an increase in c-di-GMP levels; results that were subsequently corroborated by the fluorescence measurements. Whereas no significant fluorescence was observed for inosine concentration of up to 10 μM, strong fluorescence was observed at 50 to 1000 μM (Fig. 7A). Similar observations

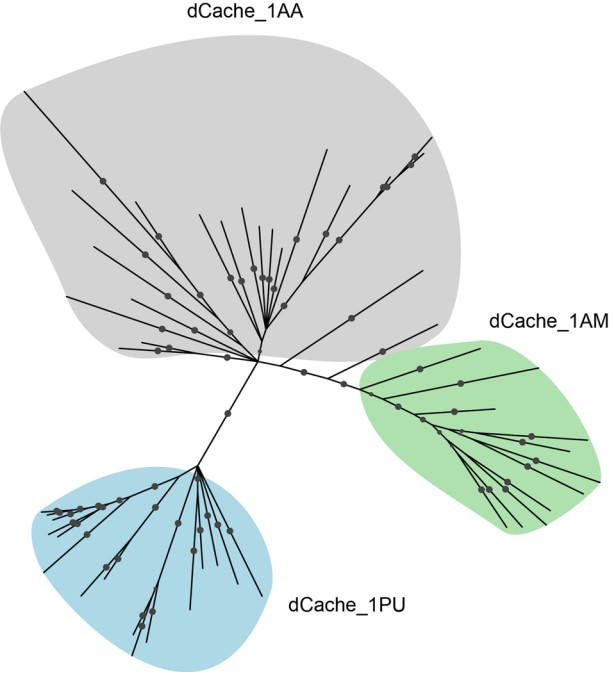

**Fig. 6 | Purine sensors share an evolutionary ancestor with amino acid sensors.** The phylogenetic tree was constructed using the Bayesian inference method implemented in MrBayes[71]. Probabilities with the value greater than 70 are shown as filled gray circles. dCache_1 amino acid sequences from major phyla of each shown family were used. The tree with information about the proteins used and phyla in which they occur can be found in Supplementary Fig. 13, the tree in NEXUS format can be found at https://github.com/ToshkaDev/Purine-pyrimidine_motif. Bayesian analyses settings are provided in Supplementary Data 3.

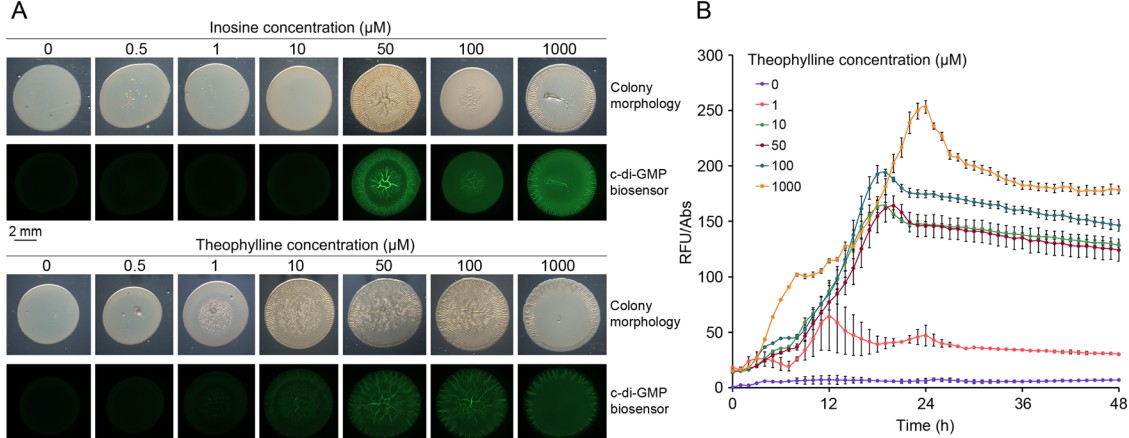

**Fig. 7 | Inosine and theophylline modulate the activity of a diguanylate cyclase (R6) of *Vibrio cholerae*. A** Colony morphology and c-di-GMP levels (green fluorescence) of *P. putida* harboring plasmid pCdrA::*gfp*^C (c-di-GMP biosensor) and pBBR1MCS-2_START_R6 (R6 expression plasmid) in the absence and presence of increasing inosine and theophylline concentrations. The brightness of the colony morphology images has been adjusted to optimize visibility. Pictures were taken after 24 h of incubation at 30 °C. **B** c-di-GMP levels during growth in liquid cultures

of *P. putida* harboring plasmid pCdrA::*gfp*^C in the absence and presence of different theophylline concentrations. Shown are means and standard deviations derived from three biological replicates each conducted in triplicate. The control experiments containing the *P. putida* harboring the empty plasmid pBBR1MCS-2_START instead of pBBR1MCS-2_START_R6 are shown in Supplementary Fig. 15. RFU relative fluorescence units, Abs absorbance at 600 nm.

have been made with theophylline. The onset of its regulatory action was well below that of inosine since the colonies started to get wrinkly at 1 µM and clear fluorescence was observed at 10 µM. Furthermore, the increase in the wrinkly morphology and fluorescence was more gradual in the case of theophylline (Fig. 7A). The differences in the regulatory activity of both compounds are likely due to the fact that inosine, but not theophylline, are metabolized by *P. putida*[22]. Similarly, guanosine also increased c-di-GMP levels and modulated the colony morphology; although the onset of the response was at 100 µM (Supplementary Fig. 14). Second, fluorescence was quantified during growth in liquid cultures in the presence of different concentrations of theophylline (Fig. 7B). Whereas almost no fluorescence was detected in the absence of theophylline, its presence at 1 µM notably augmented fluorescence levels that further increased at higher concentrations. Control experiments with the *P. putida* strain harboring the empty pBBR-based plasmid did not reveal any signal-induced changes in colony morphology or fluorescence, indicating that changes measured were due to the action of the R6 diguanylate cyclase (Supplementary Fig. 15). To consolidate the link between ligand recognition at the dCache_1PU domain and modulation of the diguanylate cyclase activity of VC2224, we have generated the N164A mutant. This residue corresponds to McpH D169 whose substitution by Ala abolished adenine binding (Table 1 and Supplementary Fig. 5). Increasing inosine concentrations did not cause any significant alteration in colony morphology or c-di-GMP in the *P. putida* strain harboring the mutant receptor (Supplementary Fig. 16). A change to the rugose colony morphology was observed at 1 mM theophylline (Supplementary Fig. 16), which is in stark contrast to the wild-type receptor where concentrations as low as 1 µM theophylline altered the colony morphology (Fig. 7). Data thus show that compounds identified as dCache_1PU domain ligands exert a regulatory function in vivo.

## Discussion

Purines and their derivatives are critical molecules required for cellular energy homeostasis and nucleotide synthesis in living organisms. In eukaryotes, purines are also important signal molecules. Purinergic receptors are expressed in almost all human tissues and organs[18,33] and are activated by the binding of purine derivatives to control processes such as cell proliferation, migration, differentiation and inflammatory responses, among others[18,34,35]. Genes encoding purinergic receptors orthologs have been identified in protozoa, algae and fungi, but are absent in bacteria[33]. However, purine derivatives were also shown to modulate a range of processes in bacteria like growth, differentiation, secondary metabolism, sporulation and adherence[33], suggesting the existence of purine-responsive receptors. To the best of our knowledge, McpH is the only so far identified bacterial receptor that recognizes purines[22].

In this study, using a recently developed iterative computational and experimental approach[11,15], we identify thousands of bacterial signal transduction proteins containing dCache_1PU domains that bind various purine derivatives. Guided by structural information, we derived and experimentally verified a defined sequence motif for this class of sensors and showed that selected dCache_1PU targets specifically bind various purine derivatives and some pyrimidines (with low affinity). dCache_1PU domains were identified in four major receptor families: chemoreceptors, histidine kinases, diguanylate cyclases and phosphodiesterases, and Ser/Thr phosphatases—receptor families that were shown to modulate functions such as chemotaxis, gene expression, second messenger turnover, and cell metabolism. We further demonstrate the physiological relevance of purine sensing using a signaling protein, which modulates second messenger (c-di-GMP) turnover in the human pathogen *V. cholerae*. Importantly, all the four receptor families mentioned above were found to play major roles in regulating bacterial virulence[36–38] and dCache_1PU domains were identified in human, animal and plant pathogenic bacteria of global relevance (Supplementary Data 2). The interference with bacterial signal transduction is an alternative strategy to fight bacterial pathogens[39–41], and the large-scale identification of signaling molecules recognized by pathogens may form the basis for developing antimicrobial agents.

The different receptor types that contain dCache_1PU domains, as well as the variety of lifestyles and ecological niches of the corresponding strains (Supplementary Data 2 and Table 2), indicate that purines are universal and omnipresent signal molecules. Interestingly, not the abundant purine or pyrimidine compounds derived from nucleic degradation, but theophylline was the ligand recognized by most of the analyzed proteins. This recognition was highly specific as the close derivative N7 methyl-theophylline (or caffeine) (Supplementary Fig. 9) was not recognized by any of the proteins analyzed (Table 3). In the literature, theophylline is generally referred to as the compound that is abundantly present in tea, coffee and chocolate. However, the inspection of metabolomics studies indicates that this compound is widespread in nature (Supplementary Table 4). It has been detected in many different organs and cells of humans, animals, plants, bacteria and different microbiomes (Supplementary Table 4). Theophylline is widely used as anti-asthma drug, since it relaxes the bronchial smooth muscle and pulmonary blood vessels; an effect caused by antagonizing the adenosine receptor[42]. The notion that theophylline is an important signal molecule is also supported by the identification of an aptamer that recognizes this molecule specifically and with high affinity ($K_D = 0.1$ µM)[43]. As in the case of dCache_1PU domains, the aptamer discriminates between theophylline and caffeine, since the affinity of the latter compound was 10,000 times reduced as compared to theophylline[43]. Theophylline binding by this aptamer also occurs naturally[44] and its specificity for theophylline has been widely exploited by synthetic biology approaches[45].

Hundreds of different LBD types exist in bacteria[2]. These domains are rapidly evolving and acquire sensing functionalities for many environmental factors[13,15,46,47]. Our analysis indicates that dCache_1PU domains shared a common ancestor with the ubiquitous amino acid sensors dCache_1AA. Given the diversity of ligand families that are recognized by dCache_1 LBDs[25], our results suggest that alternative dCache_1 subfamilies with specificity for other small molecules are likely to exist. This study demonstrates that our approach is applicable to defining ligand binding domain families at scale and reveals roles of purine derivatives as important extracellular signals in bacteria.

## Methods

### Strains and plasmids

The strains and plasmids used are specified in Supplementary Table 5. When required, 50 µg/ml kanamycin and 50 µg/ml gentamicin were added to liquid or agar medium cultures.

### Protein overexpression and purification

Wild-type and mutant derivatives of McpH-LBD were overexpressed in *E. coli* BL21 (DE3) and purified by affinity chromatography as reported previously[22]. McpH-LBD destined for crystallization trials was then dialyzed into 5 mM PIPES, 5 mM MES, 5 mM Tris, 500 mM NaCl, 10% (v/v) glycerol, pH 6.2, concentrated to about 8 mg/ml and loaded onto a HiPrep™ 26/60 Sephacryl™ S-200 size exclusion chromatography column (Cytiva, Marlborough, MA, USA) equilibrated in the same buffer. Protein was eluted isocratically with a flow of 1 ml/min. The remaining proteins were purified by affinity chromatography using the procedure used for the McpH-LBD purification, with the exception that buffers for the purification of the LBDs of receptors WP_227113344 (R8), WP_110129388 (R10) and WP_162024566 (R11) contained in addition 1 mM β-mercaptoethanol. For immediate analysis, proteins were dialyzed against the buffers provided in Supplementary Table 6. The sequences of proteins analyzed in this study are provided in Supplementary Table 7.

## Crystallization and structure resolution of McpH-LBD

Freshly purified McpH-LBD was dialyzed against 5 mM PIPES, 5 mM MES, 5 mM Tris, pH 6.2, and concentrated to 6 mg/ml using 10 kDa cut-off centricon concentrators (Amicon). Uric acid was added to the protein to a final concentration of 500 μM. Crystallization conditions were screened using the capillary counter diffusion technique[48]. Experiments were set up by placing the protein in 0.2 mm inner diameter capillaries that were then sealed with clay, and equilibrated against ad hoc precipitant cocktails designed to maximize the screening conditions[49]. Poorly diffracting crystals were obtained in 20 mM Na acetate trihydrate, 100 mM Na cacodylate trihydrate, 30 % (w/v) PEG 8 K, pH 6.5. A pH screen of this condition showed that crystals grown at pH 6.5 showed satisfactory diffraction. For data collection, crystals were equilibrated in mother solution supplemented with 15 % (v/v) glycerol, recovered with a litho-loop (Molecular Dimensions) and flash-cooled in liquid nitrogen. Data collection was done at beamlines ID23-1, ID23-2 and ID30A-1 of the European Synchrotron Radiation Facility (ESRF, Grenoble, France) and at XALOC beamline of the Spanish synchrotron radiation source Alba (Barcelona, Spain). Diffraction data from the latter radiation source were used for structure determination. Data were indexed and integrated with XDS[50], and then scaled and reduced with AIMLESS[51] of the CCP4 program suite[52]. All attempts to solve the phase problem by molecular replacement with homology models failed. However, using a model generated by AlphaFold2[23] (Supplementary Data 4, using ColabFold, generated on July 22nd, 2021), a solution was obtained using Molrep[53] and Refmac[54]. After automatic model building with Autobuild[55], refinement was continued with phenix.refine[56] of the PHENIX suite[57]. Manual re-building, water inspection and ligand identification was done in Coot[58], and final refinement was assessed including Titration-Libration-Screw (TLS) parameterization[59]. The quality of the model was verified with Molprobity[60] and the PDB validation server prior to deposition at the PDBe[61] with ID 8BMV. Supplementary Table 8 summarizes data collection and refinement statistics.

## Computational identification of dCache_1PU domains

Two iterations of PSI-BLAST search against the RefSeq database (Release 209, 2022-01-06) with a maximum number of target sequences set to 20,000 was initiated with the McpH dCache_1 domain. Subsequent iterations did not identify new sequences containing a purine motif. The identified sequences were downloaded and processed in the following way. To identify dCache_1 domain containing proteins, the downloaded sequences were scanned with the dCache_1 profile Hidden Markov Model obtained from the Pfam database (Release 35.0)[62] with the E-value threshold of 0.01 both for sequences and domains. Protein sequence regions corresponding to the dCache_1 domain were extracted and a multiple sequence alignment was built using the MAFFT (v. 7.490) FFT-NS-2 algorithm[63]. Purine motif residues were then identified and tracked on the alignment. The motif variants, domain configurations, and taxonomy information were summarized using a custom Python script. An Excel spreadsheet with the data and the multiple sequence alignment of dCache_1PU domain sequences can be found at https://github.com/ToshkaDev/Purine-pyrimidine_motif.

## Multiple sequence alignment, domain composition, and phyletic distribution tree

The multiple sequence alignment used to infer the dCache_1 domain phylogenetic tree was built using the L-INS-i algorithm of the MAFFT package (v. 7.490). Jalview v. 2.11.3.0[64] was used to explore and edit the alignments. Domains were identified running TREND (v. 1.0.0)[65,66] with the Pfam profile HMMs (v. 35.0). The generated data were downloaded in JSON format from the website and processed programmatically to determine domain architecture variants and abundances. The bacterial phylogeny for the phyletic distribution tree was retrieved from the GTDB taxonomy database (release 214)[67]. Major phyla with at least 10 genomes were depicted.

## Phylogeny inference

The multiple sequence alignment prepared to build the tree was edited using an alignment trimming tool, trimAl (v. 1.4.1)[68]: positions in the alignment with gaps in 10% or more of the sequences were removed unless this leaves less than 60%. In such case, the 60% best (with fewer gaps) positions were preserved. The amino acid replacement model for the set of protein sequences was determined running ProtTest (v. 3.4.2)[69]. The best model was found to be LG with gamma distribution of rate variation across sites in combination with the empirical state frequencies (LG + G + F). LG is an improved model compared to WAG, which has been achieved by incorporating the variability of evolutionary rates across sites in the matrix estimation and using a much larger and diverse database[70]. Using the determined amino acid replacement model, a phylogenetic tree was inferred using a Bayesian inference algorithm implemented in MrBayes[71]. Metropolis-coupled Markov chain Monte Carlo simulation implemented in MrBayes (v. 3.2.7a) was run with 3 heated and 1 cold chain and discarding the first 25% of samples from the cold chain at the "burn-in" phase. A total of 1,700,000 generations were run till the sufficient convergence was achieved (the average standard deviation of split frequencies is equal to or less than 0.01) with chain sampling every 1000 generations.

## Protein structure manipulations

Proteins were modeled using AlphaFold2[23]. Molecular docking was performed using DiffDock (https://arxiv.org/abs/2210.01776). Comparative analyses of experimental and modeled protein structures was done using PyMoL Molecular Graphics System (v. 2.5.2) and Mol* Viewer (v. 3)[72].

## Thermal shift assays

Thermal shift assays were performed using a BioRad MyiQ2 Real-Time PCR instrument. All proteins were screened against compounds of arrays PM3B (nitrogen sources) and PM5 (nutrient supplements) (Biolog, Hayward, CA, USA). Ligands were prepared by dissolving Phenotype Microarray compounds in 50 μl of Milli-Q water, that, according to the information provided by the manufacturer, corresponds to a concentration of 10 to 20 mM. In addition, solutions of ligands that were not present in the compound arrays (purine, caffeine and theophylline) were prepared in Milli-Q water at a concentration of 20 mM. Each 25 μl assay mixture contained 17–85 μM protein in their respective analysis buffer, SYPRO® Orange (Life Technologies) at 5 × concentration and 2.5 μl of the compound solution. Samples were heated from 23 °C to 85 °C at a scan rate of 0.5 °C/min. The protein unfolding curves were recorded by monitoring changes in SYPRO® Orange fluorescence. Melting temperatures were determined from the first derivative values from the raw fluorescence data.

## Isothermal titration calorimetry

Experiments were conducted on a VP-microcalorimeter (Microcal, Amherst, MA, USA) at a temperature of 25 °C. Freshly purified protein was dialyzed into the buffers detailed in Supplementary Table 6. Protein between 9 to 138 μM were placed into the sample cell and titrated with aliquots of 0.2 to 20 mM ligand solutions that were made up in dialysis buffer. Typically, a single injection of 1.6 μl was followed by a series of 4.8 to 12.8 μl aliquots. The mean enthalpies measured from the injection of ligand solutions into the buffer were subtracted from raw titration data. Data were normalized with the ligand concentrations, the first data point removed and the remaining data fitted with the 'One Binding Site' model of the MicroCal version of ORIGIN (Microcal, Amherst, MA, USA).

## Biosensor-based c-di-GMP quantification assays

Plasmid pBBR1MCS-2_START, pBBR1MCS-2_START_R6 (encoding receptor R6) or pBBR1MCS-2_START_R6 N164A (encoding the N164A mutant of R6) were transformed by electroporation into *P. putida* KT2440 harboring the reporter plasmid pCdrA::*gfp*$^C$. Overnight cultures of the resulting strains in LB medium containing the corresponding antibiotics were adjusted to an $OD_{660}$ of 1 using 1x M9 minimal medium salts. Ten µl drops were then spotted on M9 minimal medium agar plates (containing with 6 mg/l Fe-citrate, trace elements[73] and 15 mM glucose) supplemented with the appropriate antibiotics and different inosine or theophylline concentrations (0.5, 1, 10, 50, 100 and 1000 µM). Following an incubation at 30 °C for 24 h, colony morphology and fluorescence intensity were analyzed using a Leica M165 FC stereomicroscope. Fluorescence was visualized using a GFP filter (emission/excitation filter 470/525 nm). Images were taken using Leica Application Suite software using 980 ms exposure time. To monitor c-di-GMP levels during growth, 180 µl M9 minimal medium containing kanamycin and gentamicin as well as purine compounds at concentrations from 0 to 1 mM were added into 96-well black microplates (Greiner). Subsequently, 20 µl of overnight cultures diluted to an $OD_{600}$ of 0.5 in M9 minimal medium were added to each well. Growth (absorbance at 600 nm) and fluorescence (excitation at 485 nm and measurement at 520 nm) were monitored using a Varioskan LUX™ microplate reader (Thermo Scientific™). The experiments were conducted in triplicate at 30 °C under static conditions over 48 h, taking measures every hour with intermittent 10 s shaking before reading. Relative fluorescence units (RFUs) were calculated after dividing arbitrary fluorescence units by the absorbance at 600 nm to normalize fluorescence measurements with bacterial growth.

**Site-directed mutagenesis.** An overlapping PCR mutagenesis approach was employed to construct the alanine substitution mutant VC2224-N164A using primers listed in Supplementary Table 5. The resulting PCR fragment was cloned into the NdeI/BamHI sites of pBBR1MCS-2_START. The resulting plasmid was verified by PCR and DNA sequencing.

## Reporting summary

Further information on research design is available in the Nature Portfolio Reporting Summary linked to this article.

## Data availability

The model coordinates of McpH-LBD and the structure factors have been deposited at the protein data bank (https://www.rcsb.org/) with ID 8BMV. Bioinformatic data and protein models generated in this study have been deposited in the Zenodo[74]. Isothermal Titration Calorimetry binding curves have been deposited at Figshare (https://doi.org/10.6084/m9.figshare.24441052). Source data are provided with this paper.

## Code availability

Codes used have been deposited in the Zenodo database[74].

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

## Acknowledgements

This work was supported by the Spanish Ministry for Science, Innovation and Universities/*Agencia Estatal de Investigación* https://doi.org/10.13039/501100011033 (grants PID2020-112612GB-I00 to T.K., PID2019-103972GA-I00 to M.A.M. and PID2020-116261GB-I00 to J.A.G.), the Junta de Andalucía (grant P18-FR-1621 to T.K.), CSIC (grant 2023AEP002 to M.A.M.) and the NIH (grant 1R35GM131760 to I.B.Z.). We thank Mª Carmen López-Sánchez and Raquel Vázquez Santiago for technical assistance. We are grateful to the Spanish Synchrotron Light Facility (Barcelona, Spain) and the European Synchrotron Radiation Facility (Grenoble, France) for the provision of time (proposals MX1830 and Mx1938) and thank the beamline staff for their assistance with data collection.

## Author contributions

Conceptualization: I.B.Z. and T.K.; bioinformatic analyses: V.M.G.; investigation and interpretation: E.M.C., V.M.G., M.F., M.A.M., and J.A.G.; writing: I.B.Z., T.K., M.A.M., J.A.G., and V.M.G.; supervision: I.B.Z., T.K., and M.A.M.

## Competing interests

The authors declare no competing interests.
