## [Peer Review File · Nature Communications]

Ubiquitous purine sensor modulates diverse signal transduction pathways in bacteriaREVIEWER COMMENTS

Reviewer #1 (Remarks to the Author):

General comment:

The manuscript by Monteagudo-Cascales, entitled "Ubiquitous purine sensor modulates diverse signal transduction pathways in bacteria" described the presence of purine sensor in bacteria wherein in authors define the purine binding motif which present in thousands of bacterial receptors ranging from chemoreceptor, diguanylate cyclase/phosphodiesterase, histidine kinase, and serine/threonine phosphatase modulating gene expression, motility, metabolism, and recycling of secondary messengers. The hypothesis presented is intriguing, but the manuscript lacks sufficient supporting evidence and suffers from result misinterpretation. Given these shortcomings, it is not recommended for publication in this journal. The following points should be addressed to enhance the paper's quality:

Major comments

1. Since purine plays an important role in cellular processes of all living organism, its roles in bacteria are also intensively studied. Authors must modify the sentence in abstract. "However, the 26 signaling role of purines in bacteria is largely unknown"
2. There are structural interpretation errors in Figure S1 and Figure 2. Typically, at physiological pH, nitrogen atoms at positions 1, 3, 7, and 9 in purines are protonated due to the presence of carbonyl groups at 2, 6, and 8 positions. However, the authors propose that nitrogen atom at position 7 forms a hydrogen bond with Y121. As this interpretation is foundational to the authors' hypothesis, a thorough reinvestigation of the entire idea is warranted.
3. Many purine derivatives have no carbonyl group at the position 8 (except uric acid). However, authors emphasize the role of R129 as a key residue in purine binding.
4. The reviewer asserts that the overestimation of the role of key residues resulted in a misinterpretation in Figure 3. Since adenine lacks oxygen at position 8, there should be no direct interaction between R129A and adenine. Therefore, the authors need to clarify why the loss of binding activity in R129A occurs not due to the direct binding of R129 to the protein but possibly through some other indirect effect.
5. It will be informative to provide the amino acid and amine binding motif as shown for purine binding motif in Figure 2D. The sequence logo must be created for the amino acid and amine binding motif and compared with the purine binding motif to clearly see the weightage of each amino acid substitution and position in the analysis.
6. Binding studies of purine derivatives to dCache_1PU domain from various receptor families should have the negative control for the physiologically relevant primary and secondary amines and amino acids to show that these purines (amines) are specific to purine but not other amines or amino acids.
7. Can authors explain the possible reason(s) of differential binding affinities of R11 and R14 towards various substrates if they possess the identical motif (YKWFN) besides the belongs to HK and S/T kinase, respectively?
8. How would the author explain the binding of pyrimidines (cytosine and thymine) to the dCache_1PU domain, especially for R4 and R14 (Table 3)? Seeing this data can we precisely say that these are purine binding motifs? Can authors provide the atomic interpretation based on the modeling?

Minor comments:

9. Please define the PAS domain at first appearance
10. There plenty of research on adenosine signaling in bacteria naming a few (doi: 10.1080/21505594.2020.1797352, 10.1111/j.1751-7915.2012.00338.x etc), Please cite these papers in the introduction and rephrase the abstract"
11. In introduction, please rephrase or elaborate line no. 45-47.
12. Did the author include a positive control from the ITC binding experiment in the thermal shift assay to validate or correlate the binding data obtained from both the techniques?

13. In figure 4, why were R3, R7, R8, R9, R10, R12, and R13 absent. I could understand that four proteins (R3, R7, R9, R12) were insoluble, unfolded or not expressed well, but what about others (R8, R10, R13)?
14. Why did the R13 assess for the guanine binding but other purines or its derivative amines?

Reviewer #2 (Remarks to the Author):

The manuscript authored by Monteagudo-Cascales et al., describes the identification, and the structural and biochemical characterization of a novel purine sensor domain present in a variety of signaling proteins encoded in genomes of symbiotic/pathogenic bacteria and free-living bacteria. The authors also provide some insights on how the purine sensing domain may be involved in controlling c-di-GMP dependent signaling modules in the human pathogen *V. cholerae*. The manuscript is well written, it has been put together in a well-organized manner and follows a very clear logic frame. The data is explained taking into consideration a diverse audience which makes the manuscript very digestible. The question addressed by the authors is of great interest in the fields of signal transduction. I agree with the statement of the authors that one of the most difficult challenges in the field of signal transduction is the identification of ligands of physiological relevance and their receptor partners.

Comments to be addressed.

In lines 306 to 308 the authors referenced the manuscript "A novel phase variant of the cholera pathogen shows stress-adaptive cryptic transcriptomic signatures." Although I agree that it is likely that VC2224 contributes to the wrinkly phenotype of this phase variant, this was not directly addressed in the manuscript by analyzing a VC2224 deletion mutant strain. Hence, I think it could be somewhat misleading to state that "The differential expression of this gene was one of the causes for the phase variation between the rugose and smooth forms of this strain". I think it would be more appropriate to note that these gene is overexpressed in the rugose background. For instance, expression of *cdgA* and *vpsR* is induced in the rugose variants and these two have been directly involved in rugose morphology. Since *cdgA* and *vpsR* expression is regulated by c-di-GMP, their increased expression in the rugose variant could be a consequence of VC2224 induction but this was not directly tested.

The manuscript has solid experimental support for biochemical and structural based conclusions. The physiological results would be more impactful if they also validate or expand on some of the biochemical observations made with the pure proteins. I am curious if a point mutation in the asparagine residue, of the purine binding motif of VC2224, that converts it into an aspartate like in *McpH*, affects its ligand specificity or affinity and its ability to control c-di-GMP production. I think it would also be valuable to see if guanosine has a more significant effect, compared to the other purine derivatives, on the activation of VC2224. Both guanosine and inosine have been reported to accumulate in the gut in some experimental models.

The binding domain of R5 presents a very interesting case of study. This domain appears to be highly specific to hypoxanthine, another product of the gut microbiota. Furthermore, at least from the list of proteins provided in Dataset S1, R5 appears to be the only dCache_1PU bearing protein with an effector EAL domain. This *C. difficile* protein appears to have a degenerate GGDEF domain and a conserved EAL domain. Does hypoxanthine promote c-di-GMP degradation or does it inhibits PDE activity? The biochemical data also revealed that R8 and R14 and R15 are capable of binding hypoxanthine with high affinity. R5, R8, R15 and R1 have the YRWFN motif but only the first three bind hypoxanthine. How much does this 5-residue motif contributes to binding specificity? Can other specificity determinants be proposed by analyzing the sequence of the 11 proteins used in this study? I would suggest that issues of specificity of the dcache_1PU domain be addressed in the discussion section.

David Zamorano-Sánchez

Reviewer #3 (Remarks to the Author):

In this manuscript, the authors employ an impressive combination of experimental and computational approaches, identifying a widespread purine binding domain, which they name dCache_1PU, and its associated purine binding motif. It is important to note that the majority of ligand binding domains are orphans, i.e., their regulator ligands are unknown, and studies such as the one presented here are invaluable in this regard (and this one was particularly well executed). The manuscript is well presented and is a model for this type of combined experimental/computational work, which are often less enjoyable to read than this work on the purine-binding CACHE domains. The studies were nicely rounded out by demonstrating in vivo regulation of the *V. cholerae* GGDEF protein VC2224 (in a *Pseudomonas putida* system) by inosine and theophylline. The results are clearly presented, and the conclusions are solid and not overstated. The few major comments I have are easily addressable and pertain to Figure 2.

Major comments:

In Figure 2, the 2Fo-Fc density corresponding to the modeled uric acid is shown. It is more important to show the Fo-Fc difference density for this area prior to building uric acid.

In Figure 2, it appears that the 2Fo-Fc density is carved tightly around uric acid. I think it is better to show the density within a defined spherical radius, which will likely include some of the protein model and map noise as well. Perhaps it is better to leave out the 2Fo-Fc density and just show the Fo-Fc density. The way the 2Fo-Fc density is currently shown largely defeats the purpose of showing the map.

Minor comments:

Table S7 corresponds to Fig. 1. Table S7 should probably be Table S1.

Missing closing parenthesis in Table S7 "CC(1/2"

Line 147 – instead of referring to it as the "native domain" maybe refer to it as the wild-type domain or domain with wild-type sequence

Line 166 – "prefer N while others prefer D at this position" instead of "prefer," consider saying they bind more tightly with N or D at this position

Line 351 – change "single" to signal (or signaling)

Reviewer #4 (Remarks to the Author):

This study marks a significant advancement in understanding how bacteria interact with their environment through chemoreceptors. Bacteria deploy a wide array of receptors to sense and respond to environmental stimuli. Although these receptors are readily identifiable by their sequences and domain architectures, the specific ligands they bind to are largely unknown. Building on their pioneering discovery of the first bacterial chemoreceptor for purine derivatives, the authors present here the molecular basis for purine ligand binding and use this information to identify a large number of diverse chemoreceptors likely to bind purine derivatives. The study is methodically structured and delivers compelling insights. The authors first present a high-resolution crystal structure of the dCache domain of the McpH chemoreceptor, shedding light on how purine is recognized at its canonical binding site. They then use this structural information, coupled with sequence alignments, to identify a consensus motif common to many chemoreceptor ligand-binding domains. This discovery allows for the prediction of purine-binding capabilities across a diverse array of chemoreceptors. Furthermore, the team provides quantitative data on ligand affinities and specificities for a selected group of these purine-recognition domains, verifying their predictions at a broader scale. They also explore the evolutionary connections of these domains to amino acid-sensing counterparts, suggesting a shared ancestral origin. The study concludes with a practical demonstration of the regulatory effect of purine derivatives on the

activity of a diguanylate cyclase from *Vibrio cholerae*, a key representative of the identified purine chemoreceptors, in a cell-based assay.

Overall, this research represents a major leap in predicting ligands for numerous orphan chemoreceptors, employing a robust and innovative approach, offering valuable insights for both the bacterial signaling and structural biology communities.

There are a few points that would strengthen the manuscript in terms of presentation but also data analysis:

1. In the figures showing ligand binding sites in detail (i.e., Fig. 2B, Fig. 5B-F), clarity would be increased if the amino acids shown in stick presentation of the receptor would be colored according to their atom type (similar to the ligands). This adjustment would facilitate easier differentiation between polar/charged and hydrophobic residues, enhancing comprehension.

2. In Fig. 2A, the electron density of the ligand is displayed using a $|2F_o - F_c|$ map post-refinement. However, such maps can be heavily influenced by the model itself. Presenting an omit map for the ligand or a $|F_o - F_c|$ map prior to ligand inclusion would provide a more convincing representation of the experimental data, bolstering the validity of the findings.

3. The manuscript reports obtaining quantitative binding data through isothermal titration calorimetry, a method that yields both binding affinities and reaction stoichiometries. However, the stoichiometric values are conspicuously absent from the data tables and discussion. Given the application of a "one binding site model" and the crystal structure suggesting a 1:1 binding mode, the observed variability in stoichiometry (ranging from <0.1 to ~ 5 , as inferred from the titration fits) is concerning. It's essential to report these stoichiometries alongside the dissociation constants in the data tables and address this variability in the manuscript's relevant sections.

4. To further validate the role of *V. cholerae* diguanylate cyclase as a purine derivative-regulated chemoreceptor, a straightforward mutational analysis could significantly enhance the study. Creating mutant alleles that a) disrupt the ligand-binding site, or b) inactivate receptor activity would serve as valuable and appropriate controls. Such an analysis would establish more direct connections between ligand binding, receptor activity, and the observed biological effects.

RESPONSES TO REVIEWER COMMENTS

Reviewer #1 (Remarks to the Author)

The manuscript by Monteagudo-Cascales, entitled “Ubiquitous purine sensor modulates diverse signal transduction pathways in bacteria” described the presence of purine sensor in bacteria wherein in authors define the purine binding motif which present in thousands of bacterial receptors ranging from chemoreceptor, diguanylate cyclase/phosphodiesterase, histidine kinase, and serine/threonine phosphatase modulating gene expression, motility, metabolism, and recycling of secondary messengers. The hypothesis presented is intriguing, but the manuscript lacks sufficient supporting evidence and suffers from result misinterpretation. Given these shortcomings, it is not recommended for publication in this journal. The following points should be addressed to enhance the paper's quality:

Response: We thank this reviewer for his/her time and effort to review our manuscript. His/her comments were most useful to improve our manuscript. We hope to convince this reviewer that our manuscript does not suffer from result misinterpretation.

Major comments

1. Since purine plays an important role in cellular processes of all living organism, its roles in bacteria are also intensively studied. Authors must modify the sentence in abstract. “However, the 26 signaling role of purines in bacteria is largely unknown”

Response: We agree with this reviewer. The sentence in the abstract “However, the signaling role of purines in bacteria is largely unknown.” has been replaced by “In bacteria purines have also been shown to exert signaling roles.” Furthermore, several articles on the signaling role of purines in bacteria are cited. However, it has to be mentioned that the number of articles on the role of purines as signal molecules in eukaryotes is well above the corresponding number in bacteria. The sentence in the Introduction “However, their role as signal molecules in bacteria is largely unexplored.” has been replaced by “There is also evidence that purines act as signal molecules in bacteria (19–21).”

2. There are structural interpretation errors in Figure S1 and Figure 2. Typically, at physiological pH, nitrogen atoms at positions 1, 3, 7, and 9 in purines are protonated due to the presence of carbonyl groups at 2, 6, and 8 positions. However, the authors propose that nitrogen atom at position 7 forms a hydrogen bond with Y121. As this interpretation is foundational to the authors' hypothesis, a thorough reinvestigation of the entire idea is warranted.

Response: Many thanks for raising this issue. However, we would like to mention that we do not propose that the nitrogen atom at position 7 of uric acid forms a hydrogen bond with Y121. This interaction has been identified by two independent algorithms, namely Ligplot (Supplementary Fig. 1, Wallace *et al.* (1995) *Protein Engineering* 8: 127-134) and Mol* Viewer (Fig. 2B, Sehnal *et al.* (2021) *Nucleic Acids Res.* 49, W431-W437). Both programs are routinely used by the scientific community to identify binding interactions. We have also investigated the literature on the theoretical and experimental determination of the pKa value of the N7 atom of purine compounds. There appears to be a consensus in the literature (Qi *et al.* (2009) *J. Phys. Chem. B* 113, 5645-5652, Jones *et al.* (2022) *J. Phys. Chem. A* 126, 1518-1529, Mlotkowski *et al.* (2023) *J. Phys. Chem.* 127, 3526-3534, Kampf *et al.* (2002) *J. Chem. Soc., Perkin Trans. 2*, 1320-1327) that the N7 pKa of purine derivatives is between 1 to 4. This implies that N7 is deprotonated at neutral pH. For McpH-LBD crystallization, the protein in a buffer at pH 6.2 was mixed with a precipitant solution at pH 6.5, indicating that the final pH in the crystallization cocktail was in between both pH values. Under these conditions the uric acid N7 is deprotonated and able to accept a hydrogen bond from Y121. We furthermore show that Y121 is involved in ligand recognition since its replacement with alanine resulted in a 58-fold reduction in the adenine binding affinity for McpH-LBD (Fig. 3, Table 1, Supplementary Fig. 4).

3. Many purine derivatives have no carbonyl group at the position 8 (except uric acid). However, authors emphasize the role of R129 as a key residue in purine binding.

Response: Many thanks for raising this issue. A major reason for the inclusion of R129 into the consensus motif was its high conservation. The 3D structure showed that R129 of McpH-LBD interacted with the carbonyl group at position 8 of the bound ligand uric acid. Since many purine compounds do not possess a carbonyl group at position 8, we conducted docking experiments of xanthine, adenine and

purine to the three dimensional structure of McpH-LBD. These three ligands do not possess a carbonyl group at position 8, but were shown previously to bind to McpH-LBD (Fernández *et al.* (2016) *Mol Microbiol* 99:34–42). These studies show that R129 establishes hydrogen bonds with other nitrogen and oxygen atoms of the bound ligand, namely with the carbonyl groups at positions 2 and 6 of xanthine, the N1 atom of adenine and the N1 and N3 atoms of purine. We have noted that the position of these three ligands is flipped as compared to uric acid present in the 3D structure. The following sentences have been added to the Results section “R129 established a hydrogen bond with the carbonyl group at position 8 of bound uric acid (Fig. 2B). Since other McpH ligands do not possess a carbonyl group at this position, we have conducted docking experiments of further McpH ligands to the protein structure showing that R129 establishes hydrogen bonds with other parts of the bound ligand (Supplementary Fig. 5)”. Supplementary Fig. 5 (see below), has been introduced into the revised version of the manuscript.

4. The reviewer asserts that the overestimation of the role of key residues resulted in a misinterpretation in Figure 3. Since adenine lacks oxygen at position 8, there should be

no direct interaction between R129A and adenine. Therefore, the authors need to clarify why the loss of binding activity in R129A occurs not due to the direct binding of R129 to the protein but possibly through some other indirect effect.

Response: As pointed above, we have conducted docking studies of adenine and two other McpH ligands that lack a carbonyl group at position 8 to the McpH-LBD 3D structure (Supplementary Fig. 5). These experiments show that R129 establishes a hydrogen bond with the N1 atom of adenine. We hypothesize that breaking this hydrogen bond in the R129A mutant is responsible for the reduction in affinity that we observed.

5. It will be informative to provide the amino acid and amine binding motif as shown for purine binding motif in Figure 2D. The sequence logo must be created for the amino acid and amine binding motif and compared with the purine binding motif to clearly see the weightage of each amino acid substitution and position in the analysis.

Response: Many thanks for this suggestion that we have taken up. However, we prefer to show the sequence logos not in Figure 2, but as Supplementary Fig. 11 (see below) that is cited later on in the manuscript. We feel that presenting this information as part of Fig. 2 would interrupt the flow of information of this manuscript that consists in: 1) the resolution of the McpH-LBD/uric acid structure (Fig. 1); 2) the use of this structure to define the purine binding motif (Fig. 2); 3) the assessment of the weight of each amino acid of this motif in purine binding (Fig. 3); 4) the experimental identification of ligands recognized by the family of purine binding domains (Fig. 4); 5) sequence and structural comparison of the purine binding motif with the previously reported amino acid and amine binding motifs (Figs. 5 and 6); and 6) the assessment of identified purines on the activity of a full-length receptor (Fig. 7). As this reviewer indicated, this Supplementary Figure permits to assess the conservation of each of the motif residues in the three protein families.

6. Binding studies of purine derivatives to dCache_1PU domain from various receptor families should have the negative control for the physiologically relevant primary and secondary amines and amino acids to show that these purines (amines) are specific to purine but not other amines or amino acids.

Response: We agree with this reviewer. We have conducted studies to address this issue experimentally and have selected three representative proteins, namely R4, R8 and R14. These proteins were selected because they are from phylogenetically distant species (*Marinomonas spartinae* [Pseudomonadota]; *Eggerthella sinensis* [Actinomycetota]; *Sporomusa ovata* [Bacillota], respectively), belong to different receptor families (chemoreceptors, diguanylate cyclases and Ser/Thr phosphatases, respectively) and differ in the variant of the purine binding motif (FRFYN, YRWFN and YKWFN, respectively).

We have re-purified these proteins and verified by isothermal titration calorimetry that these proteins bind purine compounds. We then titrated each of the three

proteins with 15 different compounds which were found in our previous studies to bind frequently to amino acid-specific dCache domains (Gumerov *et al.* (2022) Proc. Natl. Acad. Sci. USA 119:e2110415119) and amine-specific dCache domains (Cerna-Vargas *et al.* (2023) Proc Natl Acad Sci USA 120:e2305837120).

As shown in the new Supplementary Fig. 10 (see below), all 45 titrations resulted in small and uniform heat changes that were comparable to those resulting from the injection of the compounds into buffer. We thus note an absence of binding of all compounds analyzed to these three proteins, indicating that these domains bind specifically purine compounds. This conclusion is furthermore supported by the thermal shift assays based ligand screening of the different proteins, reported in the initial version of the manuscript. As stated in the section “Computationally identified dCache_1PU domains specifically bind purine and pyrimidine derivatives”, all proteins were submitted to thermal shift ligand binding assays using the Biolog compound array PM3B, comprising nitrogen sources, and PM5, comprising nutrient supplements. These compound arrays contain an extensive range of proteinogenic and non-proteinogenic amino acids, polyamines, small amines and amides, purines, pyrimidines, amino sugars, dipeptides, inorganic ions and detergents. These compound arrays contain multiple ligands that were shown to bind to members of the amino acid and amine specific dCache_1 domains. In the corresponding results section of the initial version of this manuscript, we state that compounds that caused increases in T_m superior to 2 degrees were studied by ITC and the corresponding binding parameters are listed in Table 3. In the revised version of this manuscript, we now state explicitly that no other compounds apart from those listed in Table 3 caused increases in T_m superior to 2 degrees C.

Taken together the isothermal titration calorimetry and thermal shift assay data, we conclude that the dCache_1PU family is specific for purine compounds. Based on these new data, the following section was added to the Results part of the manuscript: “To define the ligand specificity of dCache_1PU family members, we have titrated domains R4, R8 and R14 with non-purinergetic compounds. These proteins have been selected since they form part of different receptor families, belong to phylogenetically distant species and differ in the variant of the purine binding motif (Table 2). Each protein was titrated with 15 amino acids and amines that were found to bind frequently to members of the amino acid- and amine-specific dCache_1 domains. As shown in Supplementary Fig. 10, in all cases the peaks were

small and uniform, indicative of an absence of binding of non-purinergic signals to members of the dCache_1PU family.”

II**R8****III****R14**
7. Can authors explain the possible reason(s) of differential binding affinities of R11 and R14 towards various substrates if they possess the identical motif (YKWFN) besides the belongs to HK and S/T kinase, respectively?

Response: This is an interesting issue. We have generated Alphafold2 3D models of domains R11 and R14 that were then superimposed onto the uric acid containing structure of McpH-LBD (new Supplementary Fig. 9; see below). In this superimposition all amino acids that are a distance of less than 4 Å to the bound ligand are shown in stick mode. There are several reasons that may account for differences in ligand recognition: 1) There are significant differences in the position of binding motif amino acids. This is best illustrated by R14 residue F181 that is moved by about 2 Å with respect to the corresponding residues in R11 and McpH-LBD that closely align; 2) There are differences in the amino acids that are at a distance of 4 Å to the bound ligand. This is exemplified by N149 of R11. The corresponding residue N168 in R14 is not within a distance of 4 Å from the bound ligand.

The following sentences have been added to the revised version of the manuscript: “Domains with the same motif variant differed in their ligand profiles (like R11 and R14, Table 2). These differences may be due to changes in the relative position of binding motif residues (Supplementary Fig. 9)”.

8. How would the author explain the binding of pyrimidines (cytosine and thymine) to the dCache_1PU domain, especially for R4 and R14 (Table 3)?. Seeing this data can we precisely say that these are purine binding motifs? Can authors provide the atomic interpretation based on the modeling?

Response: Thanks for raising this issue. Although some of the receptors analyzed also bound pyrimidines, we wish to refer to this domain family as purine responsive family. This is based on the following facts: 1) All proteins analyzed bound purines, whereas only four receptors bound pyrimidines. Referring to this motif as purine/pyrimidine binding motif would not be correct since only some, but not all family members bind pyrimidines; 2) For each of the four proteins that also bound pyrimidines, purines were the compounds that bound with the highest affinity, indicative of a preferential recognition. In fact, the affinities of the purine compounds that bound most tightly to these 5 proteins were 141-, 5-, 3-, 111- and 25-fold superior to those of the tightest binding pyrimidine, indicative of a clear ligand preference for purines.

To explore the structural reasons for the pyrimidine binding, we have generated an AlphaFold2 model of R4. This receptor binds preferentially purine and theophylline, but also recognizes cytosine and thymine. Purines are composed of a pyrimidine and imidazole ring. We have then docked theophylline, cytosine and thymine to the R4

model. The superimposition of these three ligands (new Supplementary Fig. 7, see below) showed that the two pyrimidine compounds overlap with the pyrimidine ring of bound theophylline, whereas the theophylline imidazole ring sticks out. The structural reason for the pyrimidine binding is thus based on the similar recognition of pyrimidines and the pyrimidine moiety of purines. The interactions that are established between the imidazole moiety of purine and the protein are thus likely to account for the in general higher affinity of purines as compared to pyrimidines. In the revised version of the manuscript the sentence “This indicates that this domain family is specific for purines and pyrimidines, which is most likely due to the fact that purines are composed of a pyrimidine and imidazole moiety (Fig. S5).” has been replaced by “This indicates that this domain family is specific for purines and pyrimidines. Purines are composed of a pyrimidine and imidazole moiety. Ligand docking studies to a R4 model showed that bound pyrimidines overlapped with the pyrimidine moiety of theophylline (Supplementary Fig. 7). The structural resemblance of pyrimidines with part of the purine structure is thus likely the reason for the pyrimidine recognition by some family members.”

Minor comments:

9. Please define the PAS domain at first appearance

Response: PAS as well as Cache were defined as Per-Arnt-Sim and Calcium channels-chemotaxis domains, respectively, on its first appearance.

10. There plenty of research on adenosine signaling in bacteria naming a few (doi: 10.1080/21505594.2020.1797352, 10.1111/j.1751-7915.2012.00338.x etc), Please cite these papers in the introduction and rephrase the abstract”

Response: The sentence in the abstract “However, the signaling role of purines in bacteria is largely unknown.” has been replaced by “In bacteria purines have also been shown to exert signaling roles.” Furthermore, several articles on the signaling role of purines in bacteria are cited. However, it has to be mentioned that the number of articles on the role of purines as signal in eukaryotes is far above the corresponding number in bacteria. In this regard, the sentence in the Introduction “However, their role as signal molecules in bacteria is largely unexplored.” has been replaced by “There is also evidence that purines act as signal molecules in bacteria”.

11. In introduction, please rephrase or elaborate line no. 45-47.

Response: We agree. The sentence “For example, PAS domains comprise one of the largest superfamilies of intracellular sensors, whereas homologous Cache domains form one of the largest superfamilies of extracellular sensors” has been changed to “PAS (Per-Arnt-Sim) and Cache (Calcium channels-chemotaxis) domains form two large superfamilies of intracellular and extracellular sensors, respectively, that are found in bacterial, archaeal and eukaryotic signal transduction systems”.

12. Did the author include a positive control from the ITC binding experiment in the thermal shift assay to validate or correlate the binding data obtained from both the techniques?

Response: We agree with the reviewer that this issue has to be better documented. In the revised version of this manuscript we now state in the section “Computationally identified dCache_1PU domains specifically bind purine and pyrimidine derivatives” that “No compounds other than those listed in Table 3 caused T_m shift superior to 2 °C”. For the very large majority of proteins that showed increases in T_m superior to 2 °C, we were able to obtain the binding

parameters using isothermal titration calorimetry. This shows that T_m shifts superior to 2 °C are indicative of binding.

We have taken up the suggestion of this reviewer and have assessed whether the failure to observe binding by ITC coincides with the failure to observe significant thermal shifts. As mentioned above, we have selected three representative domains namely R4, R8 and R14. These proteins are from phylogenetically distant species (*Marinomonas spartinae* [Pseudomonadota]; *Eggerthella sinensis* [Actinomycetota]; *Sporomusa ovata* [Bacillota], respectively), belong to different receptor families (chemoreceptors, diguanylate cyclases and Ser/Thr phosphatases, respectively) and differ in the variant of the purine binding motif (FRFYN, YRWFN and YKWFN, respectively). We have re-purified these proteins and verified by isothermal titration calorimetry that these protein bind purine compounds (Supplementary Fig. 10). We then titrated each of the three proteins with 15 different ligands. These ligands were compounds that bound frequently to members of the amino acid specific (Gumerov *et al.* (2022) Proc. Natl. Acad. Sci. USA 119:e2110415119) and amine specific dCache_1 domains (Cerna-Vargas *et al.* (2023) Proc Natl Acad Sci USA 120:e2305837120).

As shown in Supplementary Fig. 10, all 45 titrations resulted in small and uniform heat changes that were comparable to the heat changes resulting from the injection of ligand into buffer. We thus note an absence of binding of all compounds analyzed to these three proteins. All members of the family dCache_1PU have been submitted to thermal shift assays using Biolog compound arrays PM3B and PM5. Importantly, of the 15 compounds that did not show any binding using ITC to the three proteins analyzed, 11 (namely L-Val, L-Pro, L-Trp, L-Ser, L-Glu, L-Gln, L-Asp, L-Arg, choline, ethanolamine and methylamine) form part of the PM3B and PM5 compound arrays and did not produce any significant shift in the thermal shift assays, indicative of a coherence between ligand binding studies by isothermal titration calorimetry and thermal shift assays.

13. In figure 4, why were R3, R7, R8, R9, R10, R12, and R13 absent. I could understand that four proteins (R3, R7, R9, R12) were insoluble, unfolded or not expressed well, but what about others (R8, R10, R13)?

Response: We agree, the clarity of this issue needs to be improved. Of the 15 proteins that we have studied, four (R3, R7, R9, R12) were insoluble, unfolded or did not

express. Thermal shift assays of the remaining 11 proteins showed in all cases purine/pyrimidine-mediated increases in T_m above 2 °C. Of these, binding parameters could be derived for 9 proteins and the corresponding data are shown in Fig. 4 and Table 3. Contrary to this reviewer comment, the binding of ligands to R8 could be studied by ITC and data are shown in Fig. 4. For the two remaining proteins, R10 and R13, no binding was observed in ITC and, as specified in footnote to Table 3, the thermal shift assay data for both proteins are provided in Supplementary Fig. 6. Purines are relatively hydrophobic compounds and as a consequence cause very important dilution heats which prevents the injection of concentrated ligand solutions (above 2 mM in many cases), and thus the detection of lower affinity binding events. This is the reason why ITC permits the detection of higher affinity binding, whereas lower affinity binding events are also detected by the thermal shift assay.

In the revised version of the manuscript the sentence “In a number of cases, very significant increases in T_m were observed, but microcalorimetric binding heats were very weak (not permitting data analysis), indicative of low affinity binding.” was changed to “For R10 and R13, very significant increases in T_m were observed (Supplementary Fig. 6), but microcalorimetric binding heats were very weak (not permitting data analysis), indicative of low affinity binding.”

14. Why did the R13 assess for the guanine binding but other purines or its derivative amines?

Response: We agree that this not very clear. To improve the clarity, we have inserted the column “guanine” into Table 3 and state for R13 “Tsa” (Binding observed in thermal shift assays) in the case of guanine and “Nts” (No binding observed in thermal shift assays) for the remaining compounds. The statement “Evidence for the binding of guanine obtained by thermal shift assays” was removed from this Table.

Reviewer #2 (Remarks to the Author):

The manuscript authored by Monteagudo-Cascales et al., describes the identification, and the structural and biochemical characterization of a novel purine sensor domain present in a variety of signaling proteins encoded in genomes of symbiotic/pathogenic bacteria and free-living bacteria. The authors also provide some insights on how the purine sensing domain may be involved in controlling c-di-GMP dependent signaling modules in the human pathogen *V. cholerae*. The manuscript is well written, it has been put together in a well-organized manner and follows a very clear logic frame. The data is explained taking into consideration a diverse audience which makes the manuscript very digestible. The question addressed by the authors is of great interest in the fields of signal transduction. I agree with the statement of the authors that one of the most difficult challenges in the field of signal transduction is the identification of ligands of physiological relevance and their receptor partners.

Response: Dear Dr. Zamorano-Sánchez. We wish to thank you for your time and effort to review our manuscript. We are grateful for the favorable assessment of our work and your comments that were helpful to improve this manuscript.

Comments to be addressed.

In lines 306 to 308 the authors referenced the manuscript “A novel phase variant of the cholera pathogen shows stress-adaptive cryptic transcriptomic signatures.” Although I agree that it is likely that VC2224 contributes to the wrinkly phenotype of this phase variant, this was not directly addressed in the manuscript by analyzing a VC2224 deletion mutant strain. Hence, I think it could be somewhat misleading to state that “The differential expression of this gene was one of the causes for the phase variation between the rugose and smooth forms of this strain”. I think it would be more appropriate to note that these gene is overexpressed in the rugose background. For instance, expression of *cdgA* and *vpsR* is induced in the rugose variants and these two have been directly involved in rugose morphology. Since *cdgA* and *vpsR* expression is regulated by c-di-GMP, their increased expression in the rugose variant could be a consequence of VC2224 induction but this was not directly tested.

Response: We agree. The sentence “The differential expression of this gene was one of the causes for the phase variation between the rugose and smooth forms of this

strain.” has been changed to ”The *vc2224* gene was upregulated in a rugose variant, which may partially account for the rugose phenotype observed.”

The manuscript has solid experimental support for biochemical and structural based conclusions. The physiological results would be more impactful if they also validate or expand on some of the biochemical observations made with the pure proteins. I am curious if a point mutation in the asparagine residue, of the purine binding motif of VC2224, that converts it into an aspartate like in McpH, affects its ligand specificity or affinity and its ability to control c-di-GMP production.

Response: This is certainly a very interesting question. In response to another reviewer, we have generated the sequence logo of the purine binding motif using all available sequences, which is shown in Supplementary Fig. 11. Interestingly, an asparagine is predominant at this position. However, it is an aspartate in McpH and the substitution of this aspartate by asparagine caused an 80-fold reduction in the adenine binding affinity. This suggests the existence of two subfamilies, containing either an asparagine or aspartate at this position.

However, the assessment of this issue needs to be the subject of follow-up studies. We were asked by the editor to present a revised version of the manuscript within 3 months and we are simply unable to respond experimentally to all concerns raised by the four reviewers.

I think it would also be valuable to see if guanosine has a more significant effect, compared to the other purine derivatives, on the activation of VC2224. Both guanosine and inosine have been reported to accumulate in the gut in some experimental models.

Response: This is an interesting issue. Since we show that VC2224 binds 3 ligands, but have only assessed *in vivo* responses for 2 ligands, we have studied colony morphology and c-di-GMP levels in response to guanosine. These data are now shown in Supplementary Fig. 13 and the corresponding controls with the empty plasmid have been added to Supplementary Fig. 14. These data show that guanosine at 100 μ M induces the rugose colony morphology and causes high c-di-GMP levels at 1 mM. The difference in the onsets of the regulatory activities of inosine, theophylline (Fig. 7) and guanosine are most likely due to differences in their degradation, causing differential alterations in their concentrations. Whereas

theophylline is not degraded, inosine and guanosine are. There may also be differences in the velocity of inosine and guanosine degradation.

The binding domain of R5 presents a very interesting case of study. This domain appears to be highly specific to hypoxanthine, another product of the gut microbiota. Furthermore, at least from the list of proteins provided in Dataset S1, R5 appears to be the only dCache_1PU bearing protein with an effector EAL domain. This *C. difficile* protein appears to have a degenerate GGDEF domain and a conserved EAL domain. Does hypoxanthine promote c-di-GMP degradation or does it inhibits PDE activity?

Response: This is also a very interesting issue that will have to be assessed in a follow-up study. The timeframe for the submission of the revised version is of three months and in this context it will be difficult to assess in a rigorous manner all comments made by the four reviewers.

The biochemical data also revealed that R8 and R14 and R15 are capable of binding hypoxanthine with high affinity. R5, R8, R15 and R1 have the YRWFN motif but only the first three bind hypoxanthine. How much does this 5-residue motif contributes to binding specificity? Can other specificity determinants be proposed by analyzing the sequence of the 11 proteins used in this study? I would suggest that issues of specificity of the dcache_1PU domain be addressed in the discussion section.

Response: This is a most interesting issue that has also been mentioned by reviewer 1. There may be two potential reasons that may account for the differences in the ligand spectrum of proteins that share the same binding motif variant:

I) Additional amino acids that participate in ligand binding: Analysis of the McpH-LBD structure with Ligplot identified 11 amino acids that interact with bound uric acid (Supplementary Fig. 1). Of these, five form part of the binding motif. As this reviewer noted, R5, R8, R15 and R1 possess the same motif but differ in their ligand spectrum. These differences in the ligand spectrum are most likely due to variations in the nature of the remaining 6 amino acids that are not part of the binding motif but that establish contacts with the bound ligand. For example, as shown in Supplementary Fig. 2, the position of L112, establishing hydrophobic interactions of uric acid with McpH-LBD is occupied by a phenylalanine in R5, R8, R15, but is a serine in R1. Another example is F152 that also establishes hydrophobic interactions

between uric acid and McpH-LBD. In R1, the equivalent amino acid is a methionine, a serine in R5 and R8, and an alanine in R15.

However, to assess in a rigorous manner the contribution of additional amino acids to confer hypoxanthine ligand specificity, a significant number of protein variants has to be purified and analyzed. In the timeframe for the resubmission of this manuscript, such an assessment is not possible.

II) Differences in the structural organization of the amino acids that form the binding motif: Next to differences in additional amino acids that contribute to ligand binding, there is also evidence for differences in the relative position of the five binding motif residues. We have assessed this issue on the example of R11 and R14 that share the same binding motif variant but that differ significantly in their ligand spectrum. Supplementary Fig. 9 shows a structural superimposition of AlphaFold2 models of the R11 and R14 domains with the uric acid containing McpH-LBD structure. Amino acids at a distance of less than 4 Å from uric acid are shown in stick mode. R11 and R14 share the same binding site motif but differ in their ligand spectrum. This superimposition shows that the relative position of binding motif amino acids differs significantly. Most notably is the shift by about 2 Å of R181 in R14 (in yellow) with respect to the corresponding residue in R11 and McpH-LBD.

David Zamorano-Sánchez

Reviewer #3 (Remarks to the Author):

In this manuscript, the authors employ an impressive combination of experimental and computational approaches, identifying a widespread purine binding domain, which they name dCache_1PU, and its associated purine binding motif. It is important to note that the majority of ligand binding domains are orphans, i.e., their regulator ligands are unknown, and studies such as the one presented here are invaluable in this regard (and this one was particularly well executed). The manuscript is well presented and is a model for this type of combined experimental/computational work, which are often less enjoyable to read than this work on the purine-binding CACHE domains. The studies were nicely rounded out by demonstrating in vivo regulation of the *V. cholerae* GGDEF protein VC2224 (in a *Pseudomonas putida* system) by inosine and theophylline. The results are clearly presented, and the conclusions are solid and not overstated. The few major comments I have are easily addressable and pertain to Figure 2.

Response: We wish to thank this reviewer for his/her time and effort to review this manuscript, as well as for the favorable assessment. Comments made were helpful to improve this manuscript.

Major comments:

In Figure 2, the 2Fo-Fc density corresponding to the modeled uric acid is shown. It is more important to show the Fo-Fc difference density for this area prior to building uric acid.

In Figure 2, it appears that the 2Fo-Fc density is carved tightly around uric acid. I think it is better to show the density within a defined spherical radius, which will likely include some of the protein model and map noise as well. Perhaps it is better to leave out the 2Fo-Fc density and just show the Fo-Fc density. The way the 2Fo-Fc density is currently shown largely defeats the purpose of showing the map.

Response: We agree with this reviewer. In the revised version of the manuscript, Fig. 2A shows the initial |Fo-Fc| difference map prior to refinement contoured at 1.5 σ level. This initial map permits the unambiguous placement of the ligand. With this re-submission I attach the Validation Summary Report of the structure. Towards the end of this report, there is a graphical representation of the model fitted to the experimental electron density, permitting the unambiguous placement of the ligand.

Minor comments:

Table S7 corresponds to Fig. 1. Table S7 should probably be Table S1.

Response: Table S7 (now Supplementary Table 8) contains technical details of data collection and refinement, and is therefore cited in the experimental section. According to the author instructions of the journal, Tables should be numbered according to the sequence with which they are cited in the text. No changes were made.

Missing closing parenthesis in Table S7 “CC(1/2”

Response: Thanks. This has been corrected.

Line 147 – instead of referring to it as the “native domain” maybe refer to it as the wild-type domain or domain with wild-type sequence

Response: Thanks. “Native” was replaced by “wild-type”.

Line 166 – “prefer N while others prefer D at this position” instead of “prefer,” consider saying they bind more tightly with N or D at this position

Response: Many thanks. The sentence “These results may point to the fine tuning of the receptors and that some compounds prefer N while others prefer D at this position.” has been changed to “These results may point to the fine tuning of the receptors and that some compounds may bind more tightly with N instead of D at this position.”

Line 351 – change “single” to signal (or signaling)

Response: Thanks. Done.

Reviewer #4 (Remarks to the Author):

This study marks a significant advancement in understanding how bacteria interact with their environment through chemoreceptors. Bacteria deploy a wide array of receptors to sense and respond to environmental stimuli. Although these receptors are readily identifiable by their sequences and domain architectures, the specific ligands they bind to are largely unknown. Building on their pioneering discovery of the first bacterial chemoreceptor for purine derivatives, the authors present here the molecular basis for purine ligand binding and use this information to identify a large number of diverse chemoreceptors likely to bind purine derivatives. The study is methodically structured and delivers compelling insights. The authors first present a high-resolution crystal structure of the dCache domain of the McpH chemoreceptor, shedding light on how purine is recognized at its canonical binding site. They then use this structural information, coupled with sequence alignments, to identify a consensus motif common to many chemoreceptor ligand-binding domains. This discovery allows for the prediction of purine-binding capabilities across a diverse array of chemoreceptors. Furthermore, the team provides quantitative data on ligand affinities and specificities for a selected group of these purine-recognition domains, verifying their predictions at a broader scale. They also explore the evolutionary connections of these domains to amino acid-sensing counterparts, suggesting a shared ancestral origin. The study concludes with a practical demonstration of the regulatory effect of purine derivatives on the activity of a diguanylate cyclase from *Vibrio cholerae*, a key representative of the identified purine chemoreceptors, in a cell-based assay.

Overall, this research represents a major leap in predicting ligands for numerous orphan chemoreceptors, employing a robust and innovative approach, offering valuable insights for both the bacterial signaling and structural biology communities.

Response: We thank this reviewer for his/her time and effort to assess this manuscript. Comments made were most helpful to improve this manuscript.

There are a few points that would strengthen the manuscript in terms of presentation but also data analysis:

1. In the figures showing ligand binding sites in detail (i.e., Fig. 2B, Fig. 5B-F), clarity would be increased if the amino acids shown in stick presentation of the receptor would be colored according to their atom type (similar to the ligands). This adjustment would

facilitate easier differentiation between polar/charged and hydrophobic residues, enhancing comprehension.

Response: We see the point of this reviewer, but beg to differ in this issue. Throughout the manuscript and the Supplementary Material, we used a color coding for amino acids of the amino acid specific domain (grey), amine specific domain (green) and purine specific domain (blue). We use this color code for sequence alignments as well as for protein structures. To guarantee an optimal clarity of presentation, we wish to maintain this color coding.

2. In Fig. 2A, the electron density of the ligand is displayed using a $|2F_o-F_c|$ map post-refinement. However, such maps can be heavily influenced by the model itself. Presenting an omit map for the ligand or a $|F_o-F_c|$ map prior to ligand inclusion would provide a more convincing representation of the experimental data, bolstering the validity of the findings.

Response: We agree with this issue that has also been raised by reviewer 3. In the revised version, the $|2F_o-F_c|$ map has been replaced by the initial $|F_o-F_c|$ difference map contoured at the 1.5σ level. With this re-submission I attach the Validation Summary Report of the structure. Towards the end of this report, there is a graphical representation of the model fitted to the experimental electron density. The electron density permits the unambiguous placement of the ligand.

3. The manuscript reports obtaining quantitative binding data through isothermal titration calorimetry, a method that yields both binding affinities and reaction stoichiometries. However, the stoichiometric values are conspicuously absent from the data tables and discussion. Given the application of a “one binding site model” and the crystal structure suggesting a 1:1 binding mode, the observed variability in stoichiometry (ranging from <0.1 to ~ 5 , as inferred from the titration fits) is concerning. It's essential to report these stoichiometries alongside the dissociation constants in the data tables and address this variability in the manuscript's relevant sections.

Response: Many thanks for raising this issue. As this reviewer noted, crystal structures and analytical ultracentrifugation studies clearly indicate that the stoichiometry of ligand binding to dCache_1 domains is 1:1. As correctly spotted by the reviewer, in most cases the n-value is well below the expected value of 1. In the

revised version of this manuscript the n-values of the ITC data are presented in Supplementary Table 3.

We have assessed the issue of low n-values experimentally. Proteins R4, R8 and R14 have been re-purified and freshly dialyzed protein was titrated with theophylline and hypoxanthine (Supplementary Fig. 10). In all cases low n-values, comparable to those reported initially, were observed. However, titration of each of these proteins with 15 non-purine ligands resulted in each case in an absence of binding, demonstrating the purine specificity of the domains studied.

We list below the potential reasons that may account for unusually low n-values:

1. **Impure protein**: Purified protein was analyzed by SDS-PAGE. In all cases protein purity was superior to 90 % and in most cases superior to 95 %. Representative images of SDS-PAGE gels of proteins (R8 and R4) analyzed in this study are shown below.

2. **Inactive protein**: All proteins analyzed were freshly produced, i.e. purified the day before and dialyzed overnight. Proteins were never frozen. Protein purification and dialysis were done at 4 °C. Isothermal titration calorimetry experiments were conducted at a temperature that was well below the T_m values as determined by the thermal shift assays.
3. **Incorrect ligand concentrations**: The concentration of purified protein was determined by two different methods, namely spectrophotometrically using

extinction coefficients at 280 nm or by the Bradford assay. In general, there was a good agreement between both approaches, indicating that the protein concentration is subject to an acceptable error. Ligand solutions were typically prepared by weighing in solid compounds using a precision balance.

4. **Work with protein fragments:** Over the last 3 decades, the corresponding author of this manuscript has conducted ITC studies to analyze ligand binding to numerous proteins, and, mainly over the last 15 years, to protein fragments, such as ligand binding domains of bacterial receptors. When analyzing entire proteins, he noticed that n-values agree typically with the expected stoichiometry. In contrast, he realized that frequently the n-values derived from the analysis of protein fragments are well below the expected values. For example, we have observed previously low n-values for many ligand binding domains, such as: 1) a significant number of members of the dCache_1AA domain family (Fig. 2 in Gumerov *et al.* (2022) PNAS 119:e2110415119), the R7 protein (Fig. 6 in Cerna-Vargas *et al.* (2023) PNAS 120, 2305837120), PctA-LBD (Fig. 5E in Xu *et al.* (2023) mBio 14:e0209923; Fig. 2 in Monteagudo-Cascales *et al.* (2022) mBio 13:e0165022, Fig. 2 in Rico-Jimenez *et al.* (2013) Mol. Microbiol. 88:1230-43), PctC-LBD (Fig. 6E in Xu *et al.* (2023) mBio 14:e0209923) or PP1228-LBD (Fig. 5 in Corral-Lugo *et al.* (2016) Env. Microbiol. 18, 3355). In addition, low n-values for protein fragments have also been observed by other research groups, as exemplified by studies of PscA-LBD (Fig. 3 in McKellar *et al.* (2015) Mol. Micro. 96, 694-707), McpT-LBD (Fig. 7 in Baaziz *et al.* (2021) J. Bacteriol. 203, e0021621) or McpV-LBD (Fig. 7 in Compton *et al.* (2018) J. Bacteriol. 200, e00519-18). The frequent observation of low n-values for protein fragments, such as ligand binding domains of chemoreceptors, suggests that there is an equilibrium between states with and without binding affinity. We consider this as the primary reason for the low n-values observed for many of the proteins that we have studied in the present study.
5. **Imprecision in deriving n-values from hyperbolic curves:** N-values can be derived with precision from sigmoid binding curves where it corresponds to the point of inflection of the curve. In contrast, the determination of n-values from hyperbolic curves is subject to an important imprecision, which is due to the fact that the n-value is extrapolated. As a consequence, a significant number of scientists working in the field of thermodynamics and microcalorimetry do only consider

n-values derived from sigmoid but not hyperbolic curves. The majority of binding curves of this study are hyperbolic (labelled with an asterisk in Supplementary Table 3) and the intrinsic error associated with n-values derived from hyperbolic curves accounts for the variation of n-values for a given protein.

4. To further validate the role of *V. cholerae* diguanylate cyclase as a purine derivative-regulated chemoreceptor, a straightforward mutational analysis could significantly enhance the study. Creating mutant alleles that a) disrupt the ligand-binding site, or b) inactivate receptor activity would serve as valuable and appropriate controls. Such an analysis would establish more direct connections between ligand binding, receptor activity, and the observed biological effects.

Response: This is an interesting issue that we have addressed experimentally. We have generated the N164A mutant of VC2224. This residue corresponds to D169 of McpH whose replacement with alanine abolished adenine binding (Table 1, Supplementary Fig. 4). These mutant gene was cloned into a pBBR-based vector and transferred into *Pseudomonas putida* containing the c-di-GMP bioreporter plasmid pCdrA::*gfp*^C in analogy to experiments with the wild type receptor (Fig. 7). As shown in Supplementary Fig. 15, increasing inosine concentrations did not cause any change in colony morphology or c-di-GMP levels. Theophylline induced the rugose morphology at a concentration of 1 mM which contrasts with the wild type receptor for which concentrations as low as 1 microM induced a change in colony morphology. Data thus demonstrate that purine recognition at dCahce_1PU domain alters receptor activity *in vivo*.

REVIEWER COMMENTS

Reviewer #1 (Remarks to the Author):

The revised manuscript has been intensively revised and corrected according to the comments provided by the reviewers. However several key points have not been well elucidated possibly due to the authors' misinterpretation. Please consider the following points.

1. Answers to the reviewer#1-2

(1) As the reviewer pointed out in the previous review, the supplementary figure 1 must be redrawn to accommodate the allowed hydrogen bonds. The statement "Both programs are routinely used by the scientific community to identify binding interactions" seems unjustified and could be misleading.

(2) The authors' assertion that "the N7 pKa of purine derivatives is between 1 to 4" appears to be based on several references. However, it is important to note that none of the cited references specifically report the N7 pKa of uric acid, which contains a carbonyl group at position 8. This carbonyl group can influence the pKa of N7, potentially resulting in a higher pKa than the typical range of 1-4 for purine derivatives. Indeed, uric acid has been reported as a diprotic acid with $pK_{a1} = 5.4$ and $pK_{a2} = 10.3$ (McCrudden, F. H. (2008) [1905]). Therefore, while it is possible that one of the nitrogens in uric acid is deprotonated at pH 6.5 (the crystallization condition mentioned), their pKa values are not likely to be lower than 4.

(3) Based on the pH conditions provided, at pH 6.5, uric acid would predominantly exist in the form of urate ions, indicating that carbon 8 (C8) cannot be double-bonded to an oxygen atom as depicted in Supplementary Figure S1. Additionally, previous research has indicated that at pH 7.0, uric acid can exist in either diketo-enol (N3) or diketo-enol (N9) forms (Nature Communication, 2023, DOI:10.1038/s41467-023-35924-3). Therefore, the interpretation provided by the authors may not be supported by the existing evidence.

2. Answers to the reviewer#1-3 & 4

Considering the potential misinterpretation in the response to question #1-2, it appears that the authors' interpretation may not be entirely supported by the current experimental evidence and docking results. Additionally, if the orientation of purine derivatives is indeed flipped as proposed by the authors, it could disrupt the hydrogen bonding network depicted in Supplementary Figure S1, thereby rendering the authors' structural interpretation invalid. Therefore, it is highly recommended to resolve the structures of additional complexes with bound purine derivatives. Given that crystallization conditions have been well optimized, these structures could be elucidated within a relatively short time period.

Reviewer #2 (Remarks to the Author):

My concerns and observations have been thoroughly and thoughtfully addressed by the authors.

Reviewer #3 (Remarks to the Author):

[No comments for authors]

Reviewer #4 (Remarks to the Author):

The authors have addresses all my comments constructively. Congratulations to this insightful study.

1. Answers to the reviewer#1-2

(1) As the reviewer pointed out in the previous review, the supplementary figure 1 must be redrawn to accommodate the allowed hydrogen bonds. The statement "Both programs are routinely used by the scientific community to identify binding interactions" seems unjustified and could be misleading.

(2) The authors' assertion that "the N7 pKa of purine derivatives is between 1 to 4" appears to be based on several references. However, it is important to note that none of the cited references specifically report the N7 pKa of uric acid, which contains a carbonyl group at position 8. This carbonyl group can influence the pKa of N7, potentially resulting in a higher pKa than the typical range of 1-4 for purine derivatives. Indeed, uric acid has been reported as a diprotic acid with pKa1 = 5.4 and pKa2 = 10.3 (McCrudden, F. H. (2008) [1905]). Therefore, while it is possible that one of the nitrogens in uric acid is deprotonated at pH 6.5 (the crystallization condition mentioned), their pKa values are not likely to be lower than 4.

Response: We see the point made by this reviewer. In our previous response to this issue, we have provided a number of publications showing that the N7 of purines is, in general, protonated at neutral pH. As this reviewer has pointed out, uric acid is an exception and its N7 is not protonated at neutral pH. However, a hydrogen bond between N7 of uric acid and McpH Y121 has been detected by the algorithm present in LigPlot that was used to prepare Supplementary Fig. 1. The assignment of this hydrogen bond is due to the fact that the tyrosine hydroxyl group can establish hydrogen bonds both, as donor and acceptor. The following publications make reference to the capacity of the Tyr OH group to act as hydrogen bond donor and acceptor: Pace *et al.* (2001) *J. Mol. Biol.* 312, 393-404 & McDonald & Thornton (1994) *J. Mol. Biol.* 238, 777-793.

(3) Based on the pH conditions provided, at pH 6.5, uric acid would predominantly exist in the form of urate ions, indicating that carbon 8 (C8) cannot be double-bonded to an oxygen atom as depicted in Supplementary Figure S1. Additionally, previous research has indicated that at pH 7.0, uric acid can exist in either diketo-enol (N3) or diketo-enol (N9) forms (Nature Communication, 2023, DOI:10.1038/s41467-023-35924-3). Therefore, the interpretation provided by the authors may not be supported by the existing evidence.

Response: The cited study shows that at pH 7 there is an equilibrium between the diketo-enol (N3) and diketo-enol (N9) tautomers. The most abundant species (89 %) was the diketo-enol (N3) form. Since pH 7 is close to the pH used for crystallization and since the oxygen is double-bonded to the C8 atom in this predominant diketo-enol (N3) species, oxygen is also depicted as doubly bound to the C8 atom of uric acid in Supplementary Fig. 1.

Uric acid is a diprotic acid. According to (Simic and Jovanovic (1989) *J. Am. Chem. Soc.* 111, 5778-5782), pKa values of 5.4 and 9.8 were determined. Very similar values have been reported by others like Wilcox *et al.* (1972) *Med. Biol. Eng.* 10, 522-531; Finlayson and Smith (1974) *J. Chem. Eng. Data* 19, 94-97 & Wang and Königsberger (1998) *Thermochim. Acta* 310, 237-242. The structural consequences of uric acid deprotonation are depicted below. These data show that at the pH used for crystallization, the oxygen is double bonded to the C8 atom of uric acid, as shown in Supplementary Fig. 1.

Modified Figure from: Benn *et al.* (2018) *Front. Med.* 5:160.

2. Answers to the reviewer#1-3 & 4

Considering the potential misinterpretation in the response to question #1-2, it appears that the authors' interpretation may not be entirely supported by the current experimental evidence and docking results. Additionally, if the orientation of purine derivatives is indeed flipped as proposed by the authors, it could disrupt the hydrogen bonding network depicted in Supplementary Figure S1, thereby rendering the authors' structural interpretation invalid. Therefore, it is highly recommended to resolve the structures of additional complexes with bound purine derivatives. Given that crystallization conditions have been well optimized, these structures could be elucidated within a relatively short time period.

Response: Thanks for this suggestions. The elucidation of the precise binding mode of pyrimidines is certainly an interesting issue. However, following the indications of the editor, this issue will not be addressed in the framework of the present study.

REVIEWERS' COMMENTS

Reviewer #1 (Remarks to the Author):

In the revised manuscript, the raised questions and concerns have not been adequately addressed. For example, the issue of protonation at N7 and its hydrogen bond with Tyr121 has not been clearly mentioned, despite this point being raised twice.

It is certain that N7 forms a hydrogen bond with Tyr121 based on the crystal structure. In addition, N7 can indeed be protonated or deprotonated depending on the situation and environment. However, to claim the role of Tyr121 in urate binding, this point must be clearly interpreted and discussed.

Overall, since the authors failed to clearly interpret the results and provide key evidence supporting their main hypothesis, I do not recommend this paper for publication.

Reviewer #4 (Remarks to the Author):

[No comments for authors]

Point-by-point response to comments made by reviewer #1

Reviewer #1:

In the revised manuscript, the raised questions and concerns have not been adequately addressed. For example, the issue of protonation at N7 and its hydrogen bond with Tyr121 has not been clearly mentioned, despite this point being raised twice. It is certain that N7 forms a hydrogen bond with Tyr121 based on the crystal structure. In addition, N7 can indeed be protonated or deprotonated depending on the situation and environment. However, to claim the role of Tyr121 in urate binding, this point must be clearly interpreted and discussed. Overall, since the authors failed to clearly interpret the results and provide key evidence supporting their main hypothesis, I do not recommend this paper for publication.

Response: We do not share this view. Taken together, there is strong evidence that Tyr121 is involved in the binding of uric acid in McpH and that homologous residues in other purine sensors are also involved in purine sensing. However, the inspection of the literature shows that the protonation state of uric acid is currently unclear. This has now been explicitly stated in the text. The following sentence has been added to the Results section “In interpreting these binding interactions it has to be taken into account that the protonation state of the N7 atom of uric acid is unclear at present.” Since the protonation state of the N7 atom is unclear, there is also uncertainty on the placement of double bonds in the uric acid structure. Such information is impossible to derive from the X-ray structure of the McpH-LBD/uric acid complex. To avoid any imprecisions double bonds have been removed from all structural representations of uric acid and new versions of Figs. 2 and 5 as well as Supplementary Fig. 2 (Formerly Supplementary Fig. 1) were prepared.